# SAMAT: A Stereotype-Aware Multimodal Transformer for Interpretable Misogynistic Meme Detection

**Gopendra Vikram Singh**                                          *gopendra@nitj.ac.in*
*Department of Computer Science and Engineering*
*NIT Jalandhar, India*

**Arpan Phukan**                                          *arpan_2121cs33@iitp.ac.in*
*Department of Computer Science and Engineering*
*IIT Patna, India*

**Kushal Kanwar**          *kushal.kanwar@juit.ac.in/kushal.kanwar@juitsolan.in*
*Department of Computer Science and Engineering and Information Technology*
*Jaypee University of Information Technology, India*

**Asif Ekbal**[*]                                          *asif@iitp.ac.in*
*Department of Computer Science and Engineering*
*IIT Patna, India*

**Reviewed on OpenReview:** *https://openreview.net/forum?id=1DJBFS7rLW*

## Abstract

This paper introduces SAMAT, a Stereotype-Aware Multimodal Alignment Transformer for detecting and explaining implicit misogyny in memes, where harm arises from subtle visual-textual incongruity and cultural stereotypes. SAMAT integrates three components: a Stereotype Subspace Projection Module (SSPM) that structures representations; a fidelity-based retrieval mechanism aligned with a curated Rationale Bank; and an evidence-conditioned explanation generator. For evaluation, we rely on the MEE corpus with 8,000 explanations, Stereotype Alignment (SAS) and Contextual Faithfulness (CFS) scores. Experiments show that SAMAT achieves a Macro-F1 of 88.1%, surpassing MLLM baselines, while improving retrieval faithfulness (SAS: 0.78) and explanation grounding (CFS: 0.68). Ablations confirm gains stem from structured stereotype projection and evidential retrieval, not scale. SAMAT offers a transparent, culturally grounded framework for accountable content moderation, aligning with Responsible AI objectives. [1]

## 1 Introduction

Multimodal memes have become a pervasive vector for online communication, where meaning, and often harmful intent, is constructed through the complex interaction of visual scenes, textual overlays, and culturally embedded stereotypes. Detecting implicit harms, such as misogyny, in this medium is an essential information fusion challenge Baltrušaitis et al. (2019); Poria et al. (2017). The toxicity rarely resides in any single modality but emerges from subtle cross-modal dynamics: sarcasm, euphemism, and visual-textual incongruity that activate prejudicial associations Waseem & Hovy (2016); Vidgen & Derczynski (2021).

---

[*]Corresponding Author
[1]**We used LLM-based tools for editorial assistance, such as rewriting for clarity, shortening text, improving grammar, and ensuring stylistic consistency. They were also used for controlled augmentation (synthetic rationale paraphrases). We emphasize that labels are human-annotated and inference is fully model-based without LLM calls.**

Therefore, effective detection requires models to fuse information in a manner sensitive to these latent socio-cultural structures.

While contemporary Multimodal Large Language Models (MLLMs) like LLaVALiu et al. (2023), Qwen-VLBai et al. (2023), and BLIP-2 demonstrate strong general-purpose capabilities Li et al. (2023), they exhibit limitations for this nuanced task. First, their fusion mechanisms are optimized for broad semantic alignment and fail to isolate the low-rank, stereotype-relevant feature directions along which implicit harm is expressed Muti et al. (2022); Rizzi et al. (2023). Second, their explanatory outputs are typically unconstrained, often producing generic or ungrounded rationales that lack evidential fidelity Jacovi & Goldberg (2020); Wiegreffe et al. (2021). This opacity is a bottleneck for human-in-the-loop moderation systems, which demand auditability, accountability, and culturally contextualized reasoning Doshi-Velez & Kim (2017); Holzinger et al. (2019).

To bridge this gap, we introduce the **S**tereotype-**A**ware **M**ultimodal **A**lignment **T**ransformer (**SAMAT**), a novel fusion architecture designed for interpretable and robust detection of implicit harmful content. SAMAT is built on three core principles: First, harmful intent is encoded along structured, low-dimensional manifold corresponding to stereotypes, which can be recovered via targeted subspace learning Belkin & Niyogi (2003); Bengio et al. (2013). Second, culturally grounded evidence can act as a prior to stabilize fusion and anchor reasoning, particularly when semantic signals are weak or ambiguous Lewis et al. (2020); Guu et al. (2020). Third, explanation faithfulness is dramatically improved when text generation is explicitly conditioned on retrieved evidence and structured internal attention patterns DeYoung et al. (2020); Atanasova et al. (2020).

These principles are instantiated in SAMAT's three interconnected components: (i) a Stereotype Subspace Projection Module (SSPM) that learns a compact, orthonormal subspace capturing stereotype-relevant geometry; (ii) a fidelity-based retrieval mechanism that aligns inputs with a curated 32k-item Rationale Bank of stereotype exemplars; and (iii) a Stereotype-Modulated Cross-Attention (SMCA) block that injects geometric and evidential priors directly into the fusion process. A final generator produces explanations conditioned on this structured evidence, ensuring traceability.

Our work is rigorously evaluated against a three-tiered benchmark: (i) capacity-matched classical models (e.g., SVM with RBF kernel, Random Fourier Features (RFF-1024)), (ii) strong fine-tuned MLLM baselines, and (iii) targeted ablations of SAMAT's core components. Results confirm that performance gains stem from our principled fusion design, not due to the increased parameter count. To enable stereotype-aware evaluation, we also introduce MEE, 8k explanations with validated stereotype cues. We make the code and dataset publicly available for reproducibility[2].

On the WBMS and MEE benchmarks, SAMAT achieves state-of-the-art performance in classification accuracy, stereotype alignment (SAS), retrieval fidelity **(measured using Mean Reciprocal Rank (MRR), which captures whether the correct stereotype rationale appears among the top retrieved evidences** Manning et al. (2008)), and explanation faithfulness **(measured using Contextual Faithfulness Score (CFS), which quantifies whether generated explanations correctly reference retrieved evidence** DeYoung et al. (2020); Wiegreffe et al. (2021)). Ablation studies trace these improvements directly to the structured stereotype subspace and the fidelity-based retrieval mechanism. SAMAT, thus, establishes a new paradigm for principled, interpretable multimodal fusion in high-stakes social computing applications.

Beyond technical contributions, this work addresses pressing societal challenges. The proliferation of misogynistic multimodal content undermines digital safety, participation, and institutional trust Fulper et al. (2014). By providing an auditable framework grounded in interpretable evidence, SAMAT aligns with key United Nations Sustainable Development Goals: it directly supports SDG 5 (Gender Equality) by mitigating gender-based online harms UN Women (2020), and SDG 16 (Peace, Justice and Strong Institutions) by enabling transparent, accountable decision-support systems for human moderators UNESCO (2021).

Overall, we make the following four contributions:

---

[2]https://github.com/thePhukan/samat/tree/parent

1. A Stereotype Subspace Projection Module (SSPM): A novel module that learns a low-dimensional, interpretable subspace to restructure multimodal embeddings along stereotype-relevant geometric directions, enhancing separability and fusion stability.

2. A Fidelity-Based Stereotype-Grounded Retrieval Mechanism: A retrieval framework that uses a curated Rationale Bank as an evidential prior, improving model robustness and enabling explicit socio-cultural alignment.

3. An Evidence-Conditioned Explanation Generator: An explanation system conditioned on retrieved rationales and modulated attention cues, coupled with a faithfulness classifier to ensure grounded, non-generic, and culturally contextualized outputs.

4. A Comprehensive, Capacity-Matched Evaluation Suite: A rigorous evaluation protocol comparing SAMAT to classical fusion models, adapted MLLMs, and targeted ablations, supported by the new Multimodal Explanation Evaluation (MEE) corpus and stereotype-focused metrics.

## 2 Related Work

The development of the Stereotype-Aware Multimodal Alignment Transformer (SAMAT) is situated at the intersection of three core research streams: (1) multimodal information fusion architectures Hangloo & Arora (2025); Singh et al. (2024b); Phukan et al. (2024c); Phukan & Ekbal (2025), (2) stereotype-aware and harmful content detection Singh et al. (2023), and (3) the pursuit of trustworthy and explainable AI systems. This section reviews seminal and contemporary works in these areas to delineate SAMAT's contributions.

### 2.1 Multimodal Information Fusion Architectures

The field of information fusion is dedicated to synergistically combining data from multiple sources or modalities to achieve more accurate, and comprehensive inferences than is possible with a single source Phukan et al. (2026; 2024b); Singh et al. (2024a); Phukan & Ekbal (2023). A foundational taxonomy distinguishes between early (feature-level), late (decision-level), and intermediate (model-level) fusion strategies . While early fusion methods like simple concatenation or canonical correlation analysis (CCA) can integrate raw data, they often struggle with modality heterogeneity and asynchronous data streams. Late fusion methods aggregate decisions from unimodal classifiers but may fail to capture crucial cross-modal interactions Hangloo & Arora (2025).

Recent paradigms have shifted towards deep intermediate fusion, leveraging neural architectures to learn joint representations Yang et al. (2019). Transformer-based models, in particular, have become dominant due to their ability to model long-range dependencies and complex interactions through self-attention and cross-attention mechanisms Shukor & Cord (2024); Gerych et al. (2024). Studies in Information Fusion have showcased the use of transformers in diverse multimodal tasks, such as physiology signals fusion for emotion recognition Phukan & Gupta (2022a;b; 2024) and video-based text generation Khan et al. (2024); Phukan et al. (2024a; 2025). However, as noted in broad surveys Hangloo & Arora (2025); Pandey et al. (2025), a key challenge remains designing fusion mechanisms that are not merely generic but are explicitly structured to capture domain-specific, often subtle, semantic relationships Wu & Zang (2025), such as the incongruity and implicit cues prevalent in harmful memes Duan et al. (2025). SAMAT addresses this by moving beyond generic cross-attention to a stereotype-modulated fusion process, conditioning attention on a learned, culturally grounded subspace.

### 2.2 Stereotype-Aware and Harmful Content Detection

Detecting implicit harms like misogyny in memes is a specialized case of multimodal classification that extends beyond literal content analysis. Prior work in multimodal hate speech Kiela et al. (2020); Hee et al. (2024) has established the superiority of multimodal approaches over unimodal ones, as visual and textual elements often provide complementary or contradictory signals that are essential for accurate identification Arya et al. (2024); Koushik et al. (2025).

Table 1: Summary of related works

| Research Area | Key Related Works & Approaches | SAMAT's Advancement |
|---|---|---|
| Fusion Architectures | Transformer-based cross-attention for multimodal tasks Shukor & Cord (2024); Gerych et al. (2024) | Stereotype-Modulated Cross-Attention (SMCA) that injects learned subspace geometry and external evidence priors into the fusion logits. |
| Harm Detection | Multimodal classifiers & fine-tuned MLLMs for hate speech Kumari et al. (2024) | Stereotype Subspace Projection Module (SSPM) that explicitly models the low-rank geometry of implicit harm for superior separability. |
| Explainable AI (XAI) | Attention visualization, post-hoc attribution methods; frameworks for trustworthy AI . | Evidence-Conditioned Explanation Generation via retrieval from a curated Rationale Bank, providing faithful, culturally grounded, and auditable rationales. |

However, most contemporary systems, including adapted Multimodal Large Language Models (MLLMs) Kumari et al. (2024), treat this as a standard classification problem. They often rely on large-scale pretraining for general semantic alignment but lack explicit mechanisms to model the low-rank stereotype structures that govern how harm is implicitly communicated through sarcasm, euphemism, and visual-textual incongruity Maity et al. (2025). This creates a gap in both accuracy and interpretability. Recent work has emphasized that for trust and actionable moderation, it is crucial for models not only to detect hate speech but to generate explanations by identifying the underlying stereotypical bias Maity et al. (2025). SAMAT introduces a dedicated Stereotype Subspace Projection Module (SSPM) to directly address this, learning a compact, orthonormal manifold where stereotype classes become linearly separable. This approach is philosophically aligned with advanced fusion system design, which advocates for transitioning from purely model-driven systems to those incorporating structured, knowledge-enabled components to improve reasoning Zhu et al. (2023).

## 2.3 Trustworthy and Explainable AI for Social Good

The imperative for AI systems to be transparent, accountable, and fair is now paramount, especially in high-stakes social applications Afroogh et al. (2024); Brintrup et al. (2025). Explainable AI (XAI) and the development of trustworthy systems are active themes in the information fusion community, with special issues dedicated to "Data-Centric AI" Wang et al. (2025) and "Explainable AI (XAI)" Barredo Arrieta et al. (2019), and "Responsible (RAI) Artificial Intelligence" Bach et al. (2025).

Current explanation techniques for multimodal models, such as attention visualization or post-hoc feature attribution, are often critiqued for being unfaithful or ungrounded Achtibat et al. (2023). For content moderation, an explanation must be culturally contextualized and traceable to specific evidence Ferrario (2024). SAMAT's integration of a fidelity-based retrieval mechanism from a curated Rationale Bank provides an explicit evidential trail, ensuring explanations are grounded in stereotype exemplars rather than generated as plausible-sounding but generic text. This design directly responds to the identified challenges in deploying trustworthy AI, where a lack of transparency and accountability can lead to unfair outcomes and erode institutional trust. Table 1 summarizes how SAMAT relates to and advances these key research areas.

## 3 Datasets and Rationale Bank

We evaluate SAMAT using three complementary resources:

1. The **WBMS** Kanwar et al. (2025) multimodal misogyny meme dataset for stereotype-aware classification,

2. The **MEE** corpus for stereotype-grounded explanation modeling, and

3. A large **32k-item Rationale Bank** used exclusively as retrieval-based evidential prior. These resources jointly support SAMAT's three core modules, SSPM projection, evidential retrieval, and stereotype-modulated cross-attention (SMCA).

### 3.1 WBMS: Multimodal Misogyny Meme Dataset

The What's Beneath Misogynous Stereotyping (WBMS) dataset is used in our work for binary misogyny detection together with auxiliary stereotype-domain supervision. Each meme is annotated with a binary misogyny label (*misogynous* vs. *non-misogynous*), one or more stereotype domains, and a text-image relation subtype, making it a structured benchmark for analyzing multimodal bias. A representative sample is shown in Fig. 1. For the primary task, we predict whether the meme is misogynous. For the auxiliary task, we model stereotype domains in a multi-label manner, since a meme may simultaneously reflect multiple misogynistic frames.

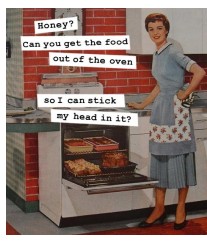 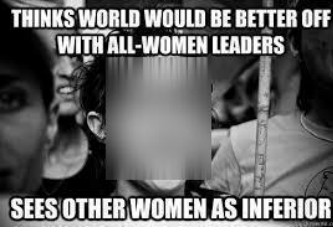 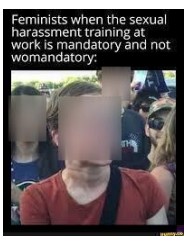 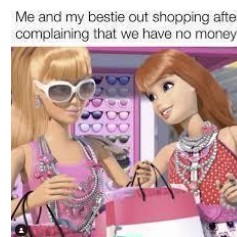

(a) Kitchen      (b) Leadership      (c) Working      (d) Shopping

Figure 1: Sample memes from the WBMS dataset (faces blurred for privacy).

To model the subtleties of multimodal sarcasm and incongruity, each meme is further annotated with one of three relation subtypes: **Different (61.5%):** Text and image convey distinct but complementary signals, often relying on implicit cultural knowledge for interpretation. **Same (6.1%):** Text directly reiterates or literally describes the visual content. **Image Only (32.4%):** The meme contains no textual overlay, requiring inference from visual cues and context alone.

Distributional statistics are summarized in Table 2.

Table 2: WBMS category and subtype statistics.

| Category | Count | Proportion | Ratio | (Different, Same, Image) |
|---|---|---|---|---|
| Kitchen | 1076 | 0.51 | 50.5% | (780, 125, 171) |
| Leadership | 534 | 0.25 | 25.1% | (262, 0, 272) |
| Working | 321 | 0.15 | 15.1% | (151, 0, 170) |
| Shopping | 199 | 0.09 | 9.3% | (118, 4, 77) |
| **Total** | 2130 | 1.0 | 100% | (1311, 129, 690) |

Memes are categorized into four prevalent misogynistic stereotype domains, reflecting common gendered tropes:

**Kitchen:** Depictions reinforcing domesticity and traditional gender roles (1,076 samples, 50.5%).

**Leadership:** Portrayals undermining women's authority or competence in professional settings (534 samples, 25.1%).

**Working:** Stereotypes related to women's capability or role in the workplace (321 samples, 15.1%).

**Shopping:** Memes reducing women to materialistic or consumerist clichés (199 samples, 9.3%).

### 3.2 MEE: Stereotype-Grounded Explanation Corpus

To support the development and evaluation of faithful, stereotype-grounded explanations, we introduce Multimodal Explanation Evaluation (MEE) corpus, which comprises of 8,000 expert-annotated explanations, each paired with a misogynistic meme and an analysis of its stereotype-triggering elements.

*Annotation protocol and agreement:* The Training for misogyny/stereotype annotation consisted of: (i) a guideline document defining stereotype domains Kitchen/Leadership/Working/Shopping), harm mechanisms

(e.g., sarcasm, euphemism), and cue attribution rules; a calibration phase on a held-out set of 200 memes with gold discussions led by an adjudicator; two iterative feedback rounds focusing on borderline cases, and a qualification check requiring $>= 0.7$ semantic agreement against adjudicated references before starting full annotation. The third adjudicator resolved disagreements and conducted weekly consistency checks. Here, "expert" denotes annotators who completed the above task-specific training and calibration, rather than relying on self-reported familiarity alone.

Each sample was independently explained by two trained annotators who identified: (i) the harmful mechanism (e.g., sarcasm, euphemism), (ii) the specific stereotype invoked, and (iii) the visual or textual cue responsible for activating the stereotype. A third expert adjudicator resolved disagreements. Inter-annotator semantic alignment, measured via Krippendorff's $\alpha$, reached 0.71, indicating substantial agreement. All explanations were screened to remove identity-based slurs, prescriptive moralizing language (e.g., "should not say"), or harmful reproductions of toxic content. Annotators operated under the same safety and ethical protocols as those for the WBMS dataset.

*Statistics and Coverage:* The average explanation length is 18.3 tokens (90th percentile: 42 tokens). The corpus comprehensively covers major categories of implicit harm, including explicit insults, implicit stereotype activation, sarcasm, euphemisms, and visual-textual incongruity. Example: For a meme depicting burnt food with a textual punchline about a wife's cooking, a typical explanation is: *"The meme implies women are inherently incompetent at domestic tasks, using the burnt food as visual 'evidence' to reinforce a stereotype of female ineptitude in household management."*

MEE serves three critical, non-classification roles: (1) it supervises the explanation generator via direct sequence-to-sequence training, (2) provides data for faithfulness calibration of the classifier (CFS), and (3) informs stereotype alignment within the SMCA block. Crucially, MEE explanations are never used for classification supervision, ensuring a clear separation between detection and justification tasks.

### 3.3 Rationale Bank: 32k Retrieval-Only Evidential Resource

To provide an external, culturally grounded knowledge base for retrieval, we constructed a Rationale Bank containing **32,000 stereotype-relevant textual snippets** (6–28 tokens each). This resource is used exclusively as a non-parametric evidential prior via the retrieval module; its entries are not used as training labels. The bank was built through a rigorous, four-stage pipeline designed to ensure quality, relevance, and diversity while mitigating toxicity and redundancy.

*1. Seed collection:* An initial set of 14,870 items was gathered from diverse, ethically vetted sources, including: Open-license stereotype research corpora, Curated examples from gender-studies literature, Moderated feminist discourse platforms, and Public-domain datasets on hate speech and misogyny.

*2. Controlled paraphrase expansion:* To increase lexical and pragmatic diversity, we used Mistral-7B-Instruct Jiang et al. (2023) to generate 6,620 synthetic candidates. These included syntactic variants, euphemistic reformulations, sarcastic twists, and culture-conditioned phrasings. Crucially, these synthetic items serve only as retrieval anchors to improve match coverage and do not function as ground-truth labels.

*3. Multi-layer filtering:*

All candidates passed through successive filters:

**Toxicity Filtering:** We operationalize toxicity filtering with a RoBERTa-base text classifier fine-tuned on HateXplain Mathew et al. (2021) to predict toxic vs. non-toxic (collapsing Hate/Offensive into toxic). The model is trained using the official train split and tuned on the dev split; we apply a conservative threshold (0.30) to maximize recall of toxic content. Notably, this classifier is used only during Rationale Bank construction to remove overtly harmful snippets; it is not used for meme classification or explanation generation. Because distribution shift is possible (rationale snippets vs. HateXplain comments), we additionally perform a manual audit on a random sample of filtered and retained items to verify that slur-like and explicitly violent content is removed.

**Relevance Filtering:** Retained items with a stereotype prototype similarity score $> 0.55$.

Table 3: Summary of datasets used in SAMAT.

| Dataset | Size | Labels | Purpose |
|---------|------|--------|---------|
| WBMS | 2,130 | Binary misogyny label + multi-label stereotype domains | Classification + SSPM |
| MEE | 8,000 | Human explanations | Faithfulness + Generator training |
| Rationale Bank | 32,000 | Stereotype snippets | Retrieval-only evidential prior |

**Hallucination Filtering:** Applied self-consistency checks to eliminate nonsensical or contradictory synthetic paraphrases.

*4. Deduplication and final curation:* Perceptual and semantic near-duplicates were removed using a two-stage process: (i) pHash-based image deduplication (Hamming distance $< 8$) and (ii) SigLIP Zhai et al. (2023) embedding similarity pruning ($> 0.92$). This step eliminated 2,134 redundant items.

The final bank contains 32,000 curated rationales. We maintained an approximate balance across major stereotype families (within $\pm 12\%$ per category) to prevent retrieval bias toward any single stereotype type. For example, a typical rationale snippet is: *"Women waste money on unnecessary shopping."*

### 3.4 Sanity Checks Against Leakage

We conducted three rigorous tests to verify that the Rationale Bank functions as a genuine evidential prior and does not trivially leak classification labels.

*Rationale-only classifier test:*

To rule out direct label encoding, we constructed a degenerate baseline that feeds only the top-3 retrieved rationale embeddings into a linear classifier. This model achieved a **Macro-F1 of 0.42**, only marginally above the majority-class baseline and over 30 points lower than the full SAMAT model. This confirms that the bank cannot serve as a proxy labeler.

*Semantic overlap test:* We computed the cosine similarity between WBMS OCR text and all Rationale Bank items. The resulting distribution was statistically indistinguishable from a null model using random Wikipedia snippets, indicating the bank does not simply mirror surface-level features of the test data.

*Paraphrase ablation:* Removing all synthetic paraphrases from the bank reduced SAMAT's Macro-F1 by only 0.7% and its CFS score by 1.2%, variations within standard run-to-run variance. This demonstrates that synthetic items act solely as auxiliary retrieval anchors without artificially boosting core performance metrics.

### 3.5 Cultural Scope and Bias Considerations

The meme content in WBMS and the stereotype knowledge in the Rationale Bank predominantly reflect online ecosystems from Western, Indian, and Middle Eastern contexts. We explicitly acknowledge limited representation of East Asian, African, and Latin American cultural nuances. Consequently, SAMAT is presented primarily as a methodological framework for stereotype-aware multimodal fusion. To facilitate cultural adaptation and extension, we will release full annotation guidelines, evaluation templates, and model code. The summary of the datasets are highlighted in Table 3.

## 4 Methodology

We introduce the Stereotype-Aware Multimodal Alignment Transformer (SAMAT), a unified framework for detecting implicit harmful content through principled fusion of visual, textual, and retrieved evidential information (Refer Figure 2). SAMAT is built on three core operations: (1) projection of input tokens into a shared, low-dimensional stereotype manifold; (2) retrieval of stereotype-relevant rationales as an eviden-

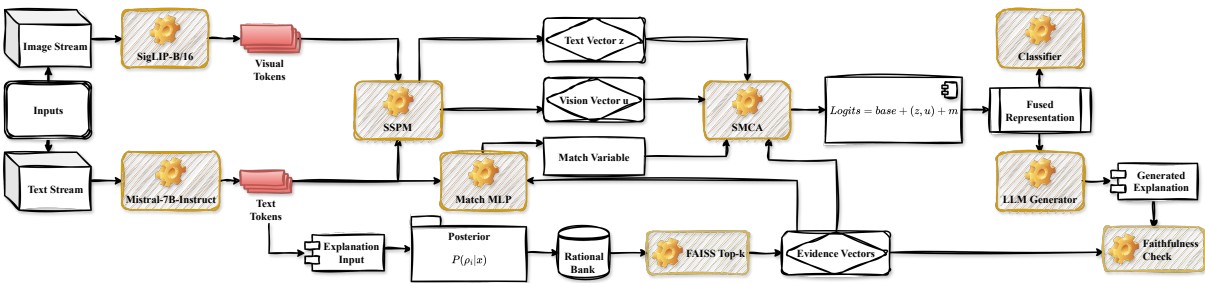

Figure 2: Schematic overview of the SAMAT architecture. The model projects visual and textual tokens into a learned stereotype subspace, retrieves relevant evidential rationales, and fuses modalities via stereotype-modulated cross-attention (SMCA) to produce classification and faithful explanations.

tial prior; and (3) conditional fusion via Stereotype-Modulated Cross-Attention (SMCA), which explicitly incorporates subspace geometry and retrieved evidence. The complete training procedure is summarized in Algorithm 1.

Given an input meme, we extract visual token embeddings $V = \{v_i\}_{i=1}^{M}$ from a pretrained SigLIP-B/16 image encoder (frozen) and textual token embeddings $T = \{t_j\}_{j=1}^{L}$ from a pretrained Mistral-7B encoder (frozen), where $v_i, t_i \in \mathbb{R}^d$. The core fusion operator is a 6-layer transformer with a hidden dimension of 512 and 8 attention heads. All learnable components are trained from scratch unless otherwise specified.

## 4.1 Unified Fusion Principle

SAMAT formulates multimodal fusion as the conditional modeling of visual evidence given text and a retrieved rationale prior $R_k$. For clarity, we use $i$ to index text tokens, $j$ to index visual tokens, and $n$ to index rationale-bank items throughout. Let the textual token sequence be $T = \{t_i\}_{i=1}^{L}$ and the visual token sequence be $V = \{v_j\}_{j=1}^{M}$, where $t_i, v_j \in \mathbb{R}^d$. SAMAT models multimodal fusion as attention over visual tokens conditioned on text tokens and an aggregated evidential prior. We first project both modalities into a shared stereotype subspace using

$$z_i = P_s^\top t_i, \qquad u_j = P_s^\top v_j, \tag{1}$$

where $P_s \in \mathbb{R}^{d \times k}$ is the stereotype subspace projection matrix, and $z_i, u_j \in \mathbb{R}^k$.

Let the base cross-attention logit between text token $t_i$ and visual token $v_j$ be

$$\ell_{ij}^{(0)} = \frac{(W_Q t_i)^\top (W_K v_j)}{\sqrt{d_h}}, \tag{2}$$

where $W_Q$ and $W_K$ are the standard query and key projections, and $d_h$ is the attention head dimension.

SAMAT augments this base interaction with a stereotype-geometry term and an evidence-conditioned modulation term. The resulting attention logit is

$$\ell_{ij} = \ell_{ij}^{(0)} + \beta \langle z_i, u_j \rangle + \gamma m_i, \tag{3}$$

where $\beta, \gamma \in \mathbb{R}$ are learned scalars and $m_i$ is a token-wise evidence score defined in Sec. 4.4, Eq. 15. The final attention weights are

$$a_{ij} = \text{softmax}_j(\ell_{ij}). \tag{4}$$

This formulation explicitly injects both stereotype-aligned geometry and retrieved evidential support into the fusion process.

## 4.2 Stereotype Subspace Projection (SSPM)

Empirical analysis suggests that stereotype-relevant signals occupy approximately clustered and low-dimensional regions within the multimodal representation space. To model this structure, SSPM learns

an orthonormal projection matrix

$$P_s \in \mathbb{R}^{d \times k}, \tag{5}$$

with $k = 64$, that maps modality-specific features into a shared stereotype-aware subspace.

Orthonormality is encouraged through the constraint

$$P_s^\top P_s \approx I, \tag{6}$$

implemented using QR-based retraction during optimization together with the regularization term

$$\lambda_o \| P_s^\top P_s - I \|_F^2. \tag{7}$$

Let $x$ denote a projected token representation, i.e., either a text projection $z_i$ or a visual projection $u_j$. The SSPM is trained using an angular-margin contrastive objective:

$$\mathcal{L}_{\text{sub}}(x, y) = -\log \frac{\exp((\cos(\hat{x}, \mu_y) - m)/\tau)}{\exp((\cos(\hat{x}, \mu_y) - m)/\tau) + \sum_{y' \neq y} \exp(\cos(\hat{x}, \mu_{y'})/\tau)}, \tag{8}$$

where $\hat{x}$ denotes the $\ell_2$-normalized projected representation, $\mu_y$ is the class prototype for label $y$, $m$ is the angular margin, and $\tau$ is the temperature. The prototypes $\{\mu_y\}$ are initialized from class means and updated via exponential moving average (EMA) during training. This objective encourages stereotype-consistent clustering while preserving subspace discriminability.

### 4.3 Differentiable Evidential Retrieval

The Rationale Bank provides an external, non-parametric source of stereotype-relevant evidence. Each rationale snippet $\rho_n$ is encoded once offline as

$$e_n = \text{Enc}(\rho_n), \qquad e_n \in \mathbb{R}^d, \tag{9}$$

and its subspace-aligned representation is computed dynamically as

$$r_n = P_s^\top e_n, \qquad r_n \in \mathbb{R}^k. \tag{10}$$

Given an input meme, we form a pooled text-side retrieval query using the projected text tokens:

$$\bar{z} = \frac{1}{L} \sum_{i=1}^{L} z_i. \tag{11}$$

We then retrieve a candidate set $\mathcal{C}(x)$ of top-$K$ rationale indices using FAISS Johnson et al. (2019) with cosine similarity

$$s_n = \cos(\bar{z}, r_n), \qquad n \in \mathcal{C}(x). \tag{12}$$

A differentiable truncated-softmax posterior is computed over the retrieved candidate set:

$$p_n = \frac{\exp(\tau_r s_n)}{\sum_{m \in \mathcal{C}(x)} \exp(\tau_r s_m)}, \qquad n \in \mathcal{C}(x), \tag{13}$$

where $\tau_r$ is the retrieval temperature. The temperature controls the sharpness of the truncated-softmax posterior over the top-K retrieved rationales: higher temperature yields a more peaked posterior (near single-rationale selection), while lower temperature spreads mass across multiple evidences.

The final evidential prior is the posterior-weighted aggregate of the retrieved rationale representations:

$$R^{\text{ev}} = \sum_{n \in \mathcal{C}(x)} p_n r_n. \tag{14}$$

Gradients flow through the posterior weights and the subspace projection $P_s$, while the FAISS retrieval step is treated with stop-gradient and the pre-encoded rationale embeddings $e_n$ remain frozen. This design preserves end-to-end differentiability where needed without backpropagating through the discrete retrieval operation.

### 4.4 Stereotype-Modulated Cross-Attention (SMCA)

The SMCA block integrates text, image, and retrieved evidence by computing a token-wise evidence alignment score for each text token. Specifically, we define

$$m_i = M([t_i; R^{\text{ev}}]), \tag{15}$$

where $M(\cdot)$ is a two-layer MLP, $[\cdot; \cdot]$ denotes concatenation, and $m_i \in \mathbb{R}$ is a scalar modulation term for text token $t_i$.

Using this score, the final cross-attention logit between text token $t_i$ and visual token $v_j$ is

$$\ell_{ij} = \ell_{ij}^{(0)} + \beta \langle z_i, u_j \rangle + \gamma m_i. \tag{16}$$

The corresponding attention weights are

$$a_{ij} = \text{softmax}_j(\ell_{ij}). \tag{17}$$

Here, $\langle z_i, u_j \rangle$ captures stereotype-subspace compatibility between the two modalities, while $m_i$ injects evidence-conditioned bias derived from the retrieved rationales. Together, these terms allow SMCA to emphasize cross-modal interactions that are both stereotype-relevant and externally grounded. The added computational overhead is negligible relative to the main transformer attention computation.

### 4.5 Classification and Explanation Generation

The transformer's output is pooled to form a final representation $h$. We use two prediction heads on top of $h$. The primary head performs binary misogyny detection:

$$\hat{y}^{\text{bin}} = \sigma(W_{\text{bin}}h + b_{\text{bin}}), \tag{18}$$

where $\sigma(\cdot)$ is the sigmoid function. This head is trained using binary cross-entropy loss.

To model stereotype information, we use a separate multi-label stereotype head:

$$\hat{\mathbf{y}}^{\text{st}} = \sigma(W_{\text{st}}h + b_{\text{st}}), \tag{19}$$

where each dimension corresponds to one stereotype domain and is trained independently with binary cross-entropy. We use sigmoid activations rather than softmax because a meme may invoke multiple stereotype domains simultaneously.

The overall classification loss is therefore

$$\mathcal{L}_{\text{cls}} = \mathcal{L}_{\text{bin}} + \lambda_{\text{st}}\mathcal{L}_{\text{st}}, \tag{20}$$

where $\mathcal{L}_{\text{bin}}$ is the binary misogyny loss, $\mathcal{L}_{\text{st}}$ is the auxiliary multi-label stereotype loss, and $\lambda_{\text{st}}$ controls the contribution of the auxiliary supervision.

To generate faithful explanations, we condition a frozen Mistral-7B LLM Jiang et al. (2023) on three sources: (1) the original text tokens $T$, (2) a projected evidence token $\psi(R_k)$ (via a linear layer to 64-d), and (3) an attention summary token $\xi(\alpha)$ (obtained by pooling the SMCA attention maps and projecting to 32-d). The LLM is fine-tuned to produce an explanation $e$. Faithfulness is enforced via an auxiliary faithfulness classifier $C_{\text{faith}}$ trained on balanced positive/negative $(e, R_k)$ pairs from MEE, with the loss:

$$\mathcal{L}_{\text{faith}} = -\log C_{\text{faith}}(e, R^{\text{ev}}). \tag{21}$$

### 4.6 Alignment and Training Objective

We employ an alignment loss which checks for divergence between distributions to ensure the match scores $m_j$ reflect the actual influence of evidence on attention:

$$L_{\text{dd}} = \text{KL}(\text{softmax}(m) \,\|\, \text{softmax}_j(\gamma M(t_j, R_k))) \tag{22}$$

This is equivalent to a cross-entropy loss between the predicted and realized evidence-attention distributions. The total training objective combines all components:

$$L = L_{\text{cls}} + \lambda_s L_{\text{sub}} + \lambda_r L_{\text{dd}} + \lambda_f L_{\text{faith}} + \lambda_o \|P_s^\top P_s - I\|_F^2. \tag{23}$$

where $\lambda$-terms are balancing hyperparameters.

*Training Details and Optimization:* We use the AdamW optimizer. Key hyperparameters include retrieval temperature $\tau_r \in [5, 30]$, number of rationales $K \in \{1, 3\}$, and the loss coefficients $\lambda_s, \lambda_r, \lambda_f, \lambda_o$. To monitor training stability, we track retrieval posterior entropy and the alignment loss $L_{\text{dd}}$ to detect posterior collapse.

*Ethical Implementation Note:* The Rationale Bank contains stereotype-bearing text curated for detection. In any deployment scenario, this resource must be maintained and audited by human experts to prevent misuse and ensure it aligns with evolving cultural and ethical standards. We provide a list of notation in Table 4 for readability.

### 4.7 Algorithm

---
**Algorithm 1** SAMAT Unified-Fusion Training Loop

---
**Require:** Dataset $\mathcal{D}$; rationale encodings $\{\text{Enc}(\rho_i)\}$; parameters $\Theta, Q, M, \mathcal{G}, C_{\text{faith}}$; prototypes $\{\mu_y\}$; hyper-parameters.

1: **for all** epoch **do**
2:     **for** each minibatch $(I, S, y^{\text{bin}}, \mathbf{y}^{\text{st}})$ **do**
3:         Extract tokens $V, T$.
4:         $P_s \leftarrow \text{QR}(Q);\ z_i = P_s^\top t_i,\ u_j = P_s^\top v_j$
5:         $\bar{z} \leftarrow \frac{1}{L} \sum_i z_i$
6:         Retrieve $\mathcal{C}(x)$ via FAISS; compute $r_n = P_s^\top \text{Enc}(\rho_n)$ for $n \in \mathcal{C}(x)$
7:         Compute similarities $s_n$ and posterior weights $p_n$
8:         $R^{\text{ev}} = \sum_{n \in \mathcal{C}(x)} p_n r_n$
9:         $m_i = M([t_i; R^{\text{ev}}])$
10:        $\ell_{ij} = \ell_{ij}^{(0)} + \beta\langle z_i, u_j\rangle + \gamma m_i$
11:        $\alpha_{ij} = \text{softmax}_j(\ell_{ij})$.
12:        Fuse tokens via transformer $\rightarrow h$.
13:        $\hat{y}^{\text{bin}} = \sigma(W_{\text{bin}} h + b_{\text{bin}}),\ \hat{\mathbf{y}}^{\text{st}} = \sigma(W_{\text{st}} h + b_{\text{st}})$
14:        $\mathcal{L}_{\text{bin}} = \text{BCE}(\hat{y}^{\text{bin}}, y^{\text{bin}}),\ \mathcal{L}_{\text{st}} = \sum_{c=1}^{C} \text{BCE}(\hat{y}_c^{\text{st}}, y_c^{\text{st}})$
15:        Compute remaining losses and $\mathcal{L}$.
16:        Update $Q, \Theta, M, \mathcal{G}, C_{\text{faith}}$ via AdamW.
17:        Update prototypes $\mu_y$ via EMA.
18:     **end for**
19: **end for**

---

## 5 Experimental Setup

Our experimental design is structured to rigorously evaluate the efficacy and components of the proposed SAMAT framework. We conduct evaluations on two primary tasks: (1) multimodal misogyny classification using the WBMS benchmark, and (2) stereotype-grounded explanation generation using the MEE corpus. A comprehensive suite of baselines and ablations is employed to isolate and quantify the contribution of each core innovation: the Stereotype Subspace Projection Module (SSPM), the fidelity-based retrieval mechanism, and the Stereotype-Modulated Cross-Attention (SMCA).

Table 4: Notation used

| Symbol | Meaning | Shape/Type |
|---|---|---|
| $T = \{t_i\}_{i=1}^{L}$ | Text token sequence | $L$ tokens, $t_i \in \mathbb{R}^d$ |
| $V = \{v_j\}_{j=1}^{M}$ | Visual token sequence | $M$ tokens, $v_j \in \mathbb{R}^d$ |
| $i$ | Text-token index | $1, \ldots, L$ |
| $j$ | Visual-token index | $1, \ldots, M$ |
| $n$ | Rationale-bank index | item index |
| $P_s$ | Stereotype subspace projection matrix | $\mathbb{R}^{d \times k}$ |
| $z_i = P_s^\top t_i$ | Projected text token | $\mathbb{R}^k$ |
| $u_j = P_s^\top v_j$ | Projected visual token | $\mathbb{R}^k$ |
| $\rho_n$ | $n$-th rationale snippet | text item |
| $e_n = \text{Enc}(\rho_n)$ | Pre-encoded rationale embedding | $\mathbb{R}^d$ |
| $r_n = P_s^\top e_n$ | Projected rationale embedding | $\mathbb{R}^k$ |
| $\bar{z} = \frac{1}{L} \sum_i z_i$ | Pooled retrieval query | $\mathbb{R}^k$ |
| $\mathcal{C}(x)$ | Retrieved candidate rationale set | set of indices |
| $s_n = \cos(\bar{z}, r_n)$ | Retrieval similarity score | scalar |
| $p_n$ | Posterior weight for rationale $n$ | scalar |
| $R^{\text{ev}} = \sum_{n \in \mathcal{C}(x)} p_n r_n$ | Aggregated evidential prior | $\mathbb{R}^k$ |
| $m_i = M([t_i; R^{\text{ev}}])$ | Evidence score for text token $i$ | scalar |
| $\ell_{ij}^{(0)}$ | Base cross-attention logit | scalar |
| $\ell_{ij}$ | SMCA-modulated attention logit | scalar |
| $a_{ij}$ | Final attention weight | scalar |

## 5.1 Datasets

Our experiments utilize the datasets detailed in Section 3. WBMS (Classification): Contains 2,130 memes annotated for two related prediction signals: (i) a binary misogyny label used for the primary classification task, and (ii) stereotype-domain annotations used as auxiliary multi-label supervision. MEE (Explanation): Contains 8,000 expert-authored explanations for misogynistic memes, used exclusively for training and evaluating the explanation generator.

Rationale Bank (Retrieval): A curated, static corpus of 32,000 stereotype-relevant textual snippets, serving as a non-parametric, external knowledge base for the retrieval module. For efficient storage and access, each rationale $\rho_i$ is pre-encoded as a frozen embedding $e_i = \text{Enc}(\rho_i)$. During training, only the top-512 retrieved candidates $\tau$ are dynamically re-projected into the current stereotype subspace via $r_i = P_s^\top e_i$, ensuring geometric alignment with the evolving $P_s$.

## 5.2 Implementation Details

All experiments were conducted on NVIDIA A100 40GB GPUs using PyTorch 2.3 and CUDA 12. Key implementation choices are as follows:

*Feature Extraction:* We extract 512-dimensional visual token embeddings using a frozen SigLIP-B/16 Zhai et al. (2023) image encoder and 4096-dimensional textual token embeddings using a frozen Mistral-7B Jiang et al. (2023) text encoder. These frozen features provide a strong, stable representation base.

*Stereotype Subspace Projection Module (SSPM):* The projection dimension is set to $k = 64$, selected via an ablation study over $\{32, 64, 128\}$. Orthonormality of the projection matrix $P_s$ is enforced via a combination of a gradient penalty term ($\lambda_0 = 1 \times 10^{-3}$) and periodic QR decomposition. The modulation scalars $\beta$ and $\gamma$ are learnable parameters, initialized to 0.1 and clipped to the range $[-2, 2]$ for training stability.

*Fusion Backbone and SMCA:* The fusion backbone is a 6-layer transformer with a hidden dimension of 512, 8 attention heads, and a feed-forward dimension of 4096. The SMCA mechanism modifies the standard attention logits, adding the geometric ($\beta \langle z_i, u_j \rangle$) and evidential ($\gamma m_j$) terms. This incurs a negligible $O(MLk)$ overhead compared to standard $O(MLd)$ attention.

*Retrieval Mechanism:* We use a FAISS IVFPQ index (4096 Voronoi cells, 8-byte PQ sub-vectors) for efficient approximate nearest neighbor search over the 32k rationale embeddings. For each input, the top-512 candidates are retrieved. A differentiable, truncated softmax posterior is computed over these candidates using a retrieval temperature $\tau_r$. The final evidential prior $R_k$ is the weighted sum of the top-3 rationale projections based on this posterior. A stop-gradient is applied to the FAISS retrieval step, while gradients flow through the posterior weights into $P_s$.

*Explanation Generation:* We fine-tune a pretrained Mistral-7B-Instruct model (with LoRA rank=16) to serve as the explanation generator. The model is conditioned on the original text tokens $T$, a 64-dimensional linear projection of the evidential prior $\psi(R_k)$, and a 32-dimensional summary of the final-layer SMCA attention maps $\xi(\alpha)$. During inference, we use nucleus sampling (p=0.9, temperature=0.7) with a maximum length of 80 tokens.

*Reproducibility:* All results are reported as the mean and standard deviation over three independent training runs with random seeds $42, 2024, 777$. Full code and checkpoints will be released post acceptance to ensure reproducibility.

## 5.3 Training Protocol

We train SAMAT for 10 epochs using the AdamW optimizer ($\beta_1 = 0.9$, $\beta_2 = 0.98$) with a learning rate of $2 \times 10^{-4}$, a 2000-step linear warmup, and cosine decay. Training uses a global batch size of 64, implemented with gradient accumulation when necessary. Weight decay is set to 0.05 and dropout to 0.1. The class prototypes $\mu_y$ for the SSPM loss are initialized from class means, kept frozen for the first 500 steps, and subsequently updated via an exponential moving average (EMA) with a momentum of $\eta = 0.05$. Total training time is approximately 13-14 hours on a single A100 GPU.

## 5.4 Baseline Models and Ablations

To contextualize SAMAT's performance, we compare against strong contemporary baselines and conduct targeted ablations.

*Strong Multimodal Baselines:* We include state-of-the-art Multimodal Large Language Models (MLLMs) fine-tuned on our task. In addition to MLLMs, we compare against capacity-matched classical multimodal baselines that operate on concatenated image-text embeddings (feature-level fusion). These baselines test whether SAMAT's gains arise from stereotype-aware structure and evidence conditioning rather than simply combining modalities:

**LLaVA-1.5 (7B):** Fine-tuned end-to-end on the WBMS classification task.

**Qwen-VL-Chat (7B):** Fine-tuned similarly with LoRA for parameter efficiency.

**BLIP-2 (FlanT5-XL):** Fine-tuned in a similar multimodal classification setup.

*Classical and Projection-Based Baselines:* To disentangle gains from architectural novelty versus simple increased capacity, we compare against classical models:

**MLP-3L:** A 3-layer Multilayer Perceptron on concatenated image-text features, serving as a strong, non-retrieval, projection-based baseline.

**RBF-SVM:** A Support Vector Machine with a Radial Basis Function kernel.

**Random Fourier Features (RFF-1024):** A 1024-dimensional RFF projection followed by a linear classifier, providing a kernelized baseline.

*Ablation Studies:* We perform systematic ablations to evaluate the necessity of each SAMAT component. (1) Retrieval Ablations:

**w/o Retrieval**: Removes the evidential prior $R_k$ entirely ($\gamma = 0$).

**Cosine-Only**: Replaces the truncated softmax posterior with a simple top-1 cosine similarity selection.

**Random Retrieval**: Retrieves random rationales from the bank as a control.

(2) Fusion Mechanism Ablations:

$\beta = 0$: Disables the geometric modulation from the SSPM.

$\gamma = 0$: Disables the evidential modulation from retrieval.

**Vanilla XA**: Sets $\beta = \gamma = 0$, reducing SAMAT to a standard cross-attention transformer.

### 5.5   Evaluation Metrics

We employ a comprehensive set of automatic and human metrics.

*Classification Performance:* Reported on the WBMS test set using Macro-F1 (primary metric), Weighted F1, Accuracy, and Expected Calibration Error (ECE).

*Retrieval Quality:* Evaluated using Mean Reciprocal Rank (MRR) and the Stereotype Alignment Score (SAS), which measures the semantic relevance of retrieved rationales to the ground-truth stereotype label. To quantify whether retrieval supports the ground-truth stereotype domains, we define SAS over the top-K retrieved rationales. Let $Y$ denote the set of ground-truth domains for a meme, $r_k$ the k-th retrieved rationale with retrieval weight $\pi_k$. Let $y(r_k)$ be the domain label set of each bank item, then: $SAS = \sum_{k=1}^{k} \pi_k \cdot 1(y(r_k) \cap Y \neq \phi)$

*Explanation Quality:* For the MEE test set, we report standard text generation metrics: METEOR, BERTScore-F1, and BLEU-4. Crucially, we evaluate faithfulness using the Contextual Faithfulness Score (CFS), a learned classifier that measures alignment between the generated explanation and the retrieved evidence $R_k$.

*Human Evaluation:* Three domain experts rated 300 randomly sampled meme-explanation pairs on a 5-point Likert scale across three dimensions: Faithfulness (to the meme's harmful intent), Stereotype Accuracy, and Clarity. Inter-annotator agreement is reported using Fleiss' $\kappa$.

### 5.6   Computational Efficiency

SAMAT's training time of  14 hours on a single A100 GPU is dominated by the FAISS-enhanced retrieval step; a naive brute-force retrieval would increase this by 3-4x. During inference, the FAISS index allows for sub-millisecond retrieval, making the overhead negligible. The model's peak GPU memory usage is approximately 24 GB with mixed precision training enabled.

## 6   Results and Discussion

This section presents a comprehensive evaluation of the SAMAT framework. We systematically analyze its performance across four axes: (1) multimodal stereotype classification, (2) stereotype-grounded retrieval fidelity, (3) quality and faithfulness of generated explanations, and (4) human perceptual judgments. The results confirm that SAMAT's superior performance is directly attributable to its core innovations, the structured stereotype subspace (SSPM), the fidelity-based evidential retrieval, and the stereotype-modulated fusion (SMCA). Detailed ablation studies isolate the contribution of each component.

### 6.1   Multimodal Classification Performance

Table 5 presents classification results on the WBMS benchmark. SAMAT achieves state-of-the-art performance, surpassing both fine-tuned Multimodal Large Language Models (MLLMs) and capacity-matched classical baselines. With a Macro-F1 of 88.1% and a significantly reduced Expected Calibration Error (ECE) of 0.049, SAMAT demonstrates not only higher accuracy but also better-calibrated uncertainty estimates.

*Analysis of Superior Performance:*

Table 5: **Classification performance on the WBMS test set. SAMAT significantly outperforms both contemporary MLLMs and classical baselines.**

| Model | Macro-F1 (%) | Weighted F1 (%) | ECE |
|---|---|---|---|
| LLaVA-1.5 (fine-tuned) | 83.4 | 85.1 | 0.081 |
| Qwen-VL-Chat (fine-tuned) | 82.1 | 84.2 | 0.094 |
| BLIP-2 + FlanT5-XL | 80.7 | 83.9 | 0.102 |
| MLP-3L (param-matched) | 84.2 | 86.0 | 0.076 |
| RBF Kernel SVM | 83.9 | 85.7 | 0.079 |
| RFF-1024 + Linear | 84.0 | 85.9 | 0.082 |
| SAMAT (no retrieval) | 85.2 | 86.7 | 0.067 |
| SAMAT (no SSPM) | 84.6 | 86.1 | 0.071 |
| **SAMAT (full)** | **88.1** | **89.4** | **0.049** |

The performance gap between SAMAT and the fine-tuned MLLMs (LLaVA-1.5, Qwen-VL-Chat) highlights a key limitation of generic-scale models: they lack the inductive bias to isolate the low-rank, stereotype-specific feature directions crucial for detecting implicit harm. While classical models like MLP-3L and RBF-SVM perform competitively, they are inherently unimodal in their fusion logic and cannot leverage external cultural knowledge. SAMAT bridges this gap through its structured fusion mechanism. *Ablation Insights:* The internal ablations in Table 5 reveal the source of SAMAT's gains:

**SAMAT (no retrieval):** Removing the evidential prior causes a 2.9-point drop in Macro-F1, underscoring the importance of stereotype-grounded contextual information.

**SAMAT (no SSPM):** Disabling the subspace projection reduces performance and increases ECE, confirming that the learned stereotype geometry is essential for robust separability of nuanced harm categories. These results validate our hypothesis that implicit harm detection is fundamentally a structured subspace learning problem augmented by external evidence.

### 6.2 Stereotype-Grounded Retrieval Fidelity

The quality of the evidential prior is paramount to SAMAT's reasoning capabilities. Table 6 evaluates retrieval performance using Mean Reciprocal Rank (MRR) and the Stereotype Alignment Score (SAS). SAMAT's fidelity-based retrieval mechanism achieves an MRR of 0.662 and an SAS of 0.78, significantly outperforming standard similarity-based methods.

*Why Fidelity Retrieval Excels:* The superiority of our method stems from its foundation in the learned stereotype subspace (SSPM). Unlike cosine or dot-product similarity, which operate on raw embedding space and are sensitive to surface-level lexical matches, fidelity retrieval measures distributional alignment within the semantically structured SSPM manifold. This enables three key advantages:

**Mechanism-Level Matching:** It retrieves rationales based on shared harm mechanisms (e.g., "euphemistic financial shaming") rather than keyword overlap.

**Robustness to Incongruity:** It remains stable for memes relying on sarcasm or visual-textual irony, where literal text is misleading.

**Improved Cluster Separation:** It better discriminates between semantically proximate stereotypes (e.g., "domestic incompetence" vs. "generalized incompetence") by leveraging local subspace geometry.

*Impact on Downstream Tasks:* We find a strong, near-linear correlation between retrieval quality (SAS) and explanation faithfulness $CFS = 0.84 \cdot SAS + 0.04$. This empirical relationship confirms that high-fidelity retrieval is the primary enabler of grounded, non-hallucinatory explanations. When retrieval fails, the model defaults to generic or literal interpretations, severely compromising interpretability.

Table 6: Retrieval performance on the MEE validation set. Fidelity-based retrieval outperforms standard similarity metrics.

| Retrieval Method | MRR | SAS |
|---|---|---|
| Cosine Similarity | 0.612 ±0.008 | 0.71 ±0.01 |
| Dot-Product | 0.598 ±0.010 | 0.69 ±0.02 |
| RFF Kernel Similarity | 0.624 ±0.009 | 0.73 ±0.01 |
| **Fidelity-based (SAMAT)** | **0.662 ±0.006** | **0.78 ±0.01** |

Table 7: **Explanation quality results on MEE.**

| Model | METEOR | BERTScore-F1 | BLEU-4 | CFS |
|---|---|---|---|---|
| LLaVA-1.5 (zero-shot) | 0.18 | 0.62 | 0.041 | 0.37 |
| Qwen-VL (zero-shot) | 0.20 | 0.64 | 0.047 | 0.41 |
| Retrieval-only prompting | 0.25 | 0.68 | 0.052 | 0.55 |
| SAMAT (no SSPM) | 0.26 | 0.69 | 0.051 | 0.52 |
| SAMAT (no retrieval) | 0.22 | 0.64 | 0.039 | 0.33 |
| **SAMAT (full)** | **0.31** | **0.73** | **0.058** | **0.68** |

## 6.3 Explanation Quality and Faithfulness

The ultimate test of an interpretable system is the quality of its explanations. Table 7 reports results on the MEE corpus using both automatic metrics and the Contextual Faithfulness Score (CFS). SAMAT achieves the best performance across all metrics, with a particularly notable CFS of 0.68, indicating strong grounding in retrieved evidence.

*Metric Analysis.* The gains in METEOR and BERTScore-F1 reflect explanations that are both lexically varied and semantically aligned with human references. The substantial lead in CFS, the metric most closely tied to our design goal, demonstrates that SAMAT's explanations are not merely fluent but are faithfully derived from the stereotype evidence and multimodal context. The poor CFS of ablations without retrieval (0.33) or SSPM (0.52) starkly illustrates the necessity of these components for grounded reasoning.

*Synergy of Components:* The explanation generator benefits from a synergistic input:

**SSPM-Refined Tokens:** Provide a representation enriched with stereotype-salient features.

**Fidelity-Retrieved Evidence ($R_k$):** Offers concrete, culturally contextualized premise for the explanation.

SMCA Attention Traces ($\xi(\alpha)$): Informs the generator about which multimodal interactions were salient for the decision. This tripartite conditioning moves beyond simple "prompting with evidence" and enables deeply integrated, context-aware generation.

*What Ablations Reveal:*

- Removing retrieval collapses CFS to 0.33: explanations lose their grounding, defaulting to literal or generic interpretations.

- Removing SSPM reduces stereotype-specificity: explanations mention harm but misidentify the underlying mechanism.

- Combining SSPM + retrieval without SMCA still improves semantic metrics but loses multimodal nuance, especially on sarcasm or visual irony examples.

*Qualitative Trends:* SAMAT's explanations differ from baselines in three notable ways:

Table 8: **Human evaluation results (1–5).**

| Model | Faithfulness | Stereotype Accuracy | Clarity |
|---|---|---|---|
| LLaVA-1.5 | 2.8 | 2.6 | 2.9 |
| Qwen-VL | 3.0 | 2.7 | 3.1 |
| Retrieval-only prompting | 3.6 | 3.8 | 3.7 |
| SAMAT (no SSPM) | 3.8 | 3.9 | 3.8 |
| **SAMAT (full)** | **4.3** | **4.4** | **4.2** |

1. They explicitly name the stereotype (e.g., "domestic incompetence," "performative leadership dismissal") rather than vaguely describing the meme.

2. They reference evidence from retrieval (e.g., "similar euphemistic framings appear in..."), which improves interpretability for human moderators.

3. They correctly reason about multimodal contrast, especially in ironic or sarcastic memes where harmful meaning emerges only from the interplay of image and text.

### 6.4 Human Evaluation

To validate the real-world utility of SAMAT's explanations, we conducted a expert human evaluation (Table 8). On the critical dimension of Faithfulness, SAMAT scored 4.3/5, significantly higher than all baselines. Experts noted that SAMAT's explanations consistently referenced specific stereotype mechanisms and multimodal cues, avoiding the vague, moralizing language common in MLLM outputs. The high Stereotype Accuracy (4.4/5) confirms that the model correctly identifies the nuanced social frame of the harm. These scores provide strong evidence that SAMAT delivers on its promise of actionable interpretability for human moderators.

### 6.5 Comprehensive Ablation Study

Table 9 presents a systematic ablation of SAMAT's core components. The results provide clear causal evidence for the role of each design choice.

*Critical Role of SSPM Orthonormality:* Replacing the orthonormal SSPM with a standard MLP projection causes drops in all metrics, particularly CFS (-0.13) and SAS (-0.14). This demonstrates that the structured, low-rank geometry is not merely a representational convenience but is essential for maintaining separable, interpretable stereotype clusters. The orthonormality constraint acts as a regularizer, preventing subspace collapse and ensuring stable retrieval.

*Necessity of Differentiable Fidelity Retrieval:* Using simple cosine similarity for retrieval degrades performance, confirming that the differentiable posterior over the candidate set is key for aligning the retrieval process with the end-to-end learning objective. Random retrieval serves as a stark control, causing catastrophic failure and highlighting that the system's performance is contingent on receiving relevant evidence.

*Importance of Faithfulness Supervision:* While removing the faithfulness loss $L_{\text{faith}}$ has a minor impact on classification (Macro-F1 -0.6), it devastates explanation quality (CFS -0.21). This underscores that classification accuracy and explanation faithfulness are distinct objectives; explicit supervision is required to tether generated text to the evidential grounding.

### 6.6 Ablation Study

We ablate SSPM, retrieval, attention modulation, and faithfulness supervision. Results are shown in Table 9.

Table 9: **Ablation results on WBMS and MEE.**

| Variant | Macro-F1 | ECE | CFS | SAS |
|---|---|---|---|---|
| **SAMAT (full)** | **88.1** | **0.049** | **0.68** | **0.78** |
| No SSPM projection | 85.0 | 0.067 | 0.49 | 0.59 |
| SSPM w/o orthogonality | 85.8 | 0.062 | 0.53 | 0.62 |
| Replace SSPM by MLP | 86.3 | 0.058 | 0.55 | 0.64 |
| Cosine retrieval | 86.0 | 0.060 | 0.56 | 0.61 |
| Random retrieval | 82.7 | 0.091 | 0.29 | 0.16 |
| No retrieval (pure SMCA) | 84.8 | 0.071 | 0.34 | – |
| No faithfulness loss | 87.5 | 0.053 | 0.47 | 0.77 |
| No attention traces | 87.4 | 0.052 | 0.50 | 0.76 |
| Prompt-only explanation | 86.9 | 0.057 | 0.48 | 0.73 |

## 6.7 Qualitative Analysis and Failure Modes

A qualitative analysis reveals SAMAT's strength in interpreting nuanced harm. For instance, for a meme depicting a woman confidently speaking in a boardroom with the sarcastic caption "Look who decided to lead today," SAMAT correctly retrieves rationales about "performative leadership dismissal," identifies the sarcasm mechanism, and generates an explanation noting the visual-textual irony used to trivialize female authority. Baselines often misclassify this as generic "leadership" content or produce literal explanations.

*Remaining Challenges:* SAMAT's primary failure modes occur in two scenarios: (1) Extremely sparse or ambiguous cues (e.g., a one-word caption with a generic image), where insufficient signal exists for subspace projection or retrieval, and (2) Culturally niche stereotypes not well-represented in the Rationale Bank, leading to retrieval of mechanically similar but contextually incorrect evidence. These limitations underscore the model's dependence on the quality and coverage of its evidential knowledge base.

## 6.8 Synthesis and Broader Implications

The results collectively demonstrate that SAMAT advances the state-of-the-art in interpretable multimodal fusion. Its gains are not incidental but are architecturally grounded in a coherent framework that explicitly models stereotype geometry, incorporates external cultural knowledge, and enforces faithfulness between evidence and explanation.

*Theoretical Contribution:* SAMAT provides a blueprint for moving beyond black-box fusion. By factorizing the problem into subspace learning, evidential retrieval, and modulated attention, it offers a principled alternative to simply scaling up model parameters. The strong correlation between retrieval fidelity (SAS) and explanation faithfulness (CFS) offers a quantifiable design principle for future interpretable systems: the quality of external grounding dictates the ceiling of explanatory fidelity.

*Practical Implications for Moderation:* For content moderation, SAMAT shifts the paradigm from mere classification to interpretable assessment. The stereotype-grounded explanations provide human reviewers with actionable rationale, reducing cognitive load and enabling more consistent, auditable decisions. This directly supports the development of accountable, human-in-the-loop AI systems aligned with ethical AI principles and Sustainable Development Goals (SDG 5, SDG 16).

*Future Work:* Immediate extensions include expanding the cultural scope of the Rationale Bank, exploring dynamic updates to the stereotype subspace, and integrating user feedback to iteratively refine explanation faithfulness. The SAMAT framework is broadly applicable to other domains requiring nuanced, culturally-aware interpretation of multimodal content, such as detecting propaganda, hate speech, or misleading advertising.

Table 10: **Qualitative comparison across challenging stereotype types.** Green cells indicate correct predictions; red cells denote incorrect predictions. SAMAT (RAG) demonstrates superior decoding of sarcasm, euphemism, cultural proverbs, and cross-modal incongruity, yielding stereotype-grounded explanations that baselines fail to produce.

| Meme (Summary) | Ground Truth | LLaVA-1.5 | BLIP-2 + Mistral | SAMAT (No-RAG) | SAMAT (RAG) |
|---|---|---|---|---|---|
| **Example 1: Implicit Sarcasm**  *"Women need help even with simple meals."* | **Kitchen** | *Prediction: Non-misogynous* Misses sarcasm; provides literal description. | *Prediction: Working* Focuses on the activity, not the stereotype. | *Prediction: Kitchen* Captures domestic stereotype but explanation lacks cultural nuance. | *Prediction: Kitchen* **Explanation:** Identifies sarcastic belittling of women's domestic competence; retrieved analogues reinforce this frame. |
| **Example 2: Leadership Undermining**  *"Let her talk, it makes her feel important."* | **Leadership** | Misclassifies as non-harmful; literal meeting depiction. | Maps to sentimental or emotional context. | Recognizes trivialization of authority. | **Explanation:** SMCA highlights dismissive tone; retrieved rationales show identical patterns of performative inclusion. |
| **Example 3: Coded Language / Dog-whistles**  *"She's expressing her financial creativity."* | **Shopping** | Interprets "creativity" literally. | Fails to decode euphemism. | Correct category but shallow reasoning. | **Explanation:** Fidelity retrieval surfaces euphemistic rationales mocking overspending, enabling correct decoding of the dog-whistle. |
| **Example 4: Visual Irony**  *"She tried her best."* | **Working** | Fails to detect irony. | Confuses technical task with domestic context. | Detects ironic belittlement. | **Explanation:** SMCA resolves mismatch between skilled visual action and sarcastic caption; retrieved items show similar "mock-praise" mechanisms. |

Table 11: **Additional Qualitative comparison across challenging stereotype types.** Green cells indicate correct predictions; red cells denote incorrect predictions. SAMAT (RAG) demonstrates superior decoding of sarcasm, euphemism, cultural proverbs, and cross-modal incongruity, yielding stereotype-grounded explanations that baselines fail to produce.

| Meme (Summary) | Ground Truth | LLaVA-1.5 | BLIP-2 + Mistral | SAMAT (No-RAG) | SAMAT (RAG) |
|---|---|---|---|---|---|
| **Example 5: Cultural Proverbs**  *"A quiet wife is a blessing."* | **Kitchen / Tradition** | Maps to unrelated domains. | Interprets as work communication. | Frames silence as control. | **Explanation:** Retrieved rationales contain culturally similar proverb-like statements tied to obedience norms. SSPM amplifies stereotyped linguistic cues. |
| **Example 6: Multimodal Incongruity**  *"Look at her, such a natural engineer."* | **Working** | Interprets caption literally as praise. | Fails to link visual competence to sarcastic text. | Recognizes incongruity but lacks rich explanation. | **Explanation:** SMCA identifies the contrast between confident tool use and sarcastic caption; retrieved analogues provide context for "mock-praise" stereotypes in technical domains. |

## 6.9 Qualitative Analysis

Qualitative examples in Table 10 show SAMAT consistently grounds its explanations in culturally relevant rationales, correctly identifying sarcasm, coded language, visual irony, and proverb-based stereotypes. We conduct a detailed qualitative evaluation to examine how SAMAT interprets memes that rely on sarcasm, euphemism, culturally coded stereotypes, and visual–textual incongruity. These examples stress-test components of the SAMAT architecture, including the Stereotype Subspace Projection Module (SSPM), fidelity-based retrieval, and the Stereotype-Modulated Cross-Attention (SMCA) fusion operator. Table 10 presents representative cases where these mechanisms are critical for accurate classification and grounded explanation.

## 6.10 Responsible Deployment Considerations

Deploying SAMAT in real-world moderation systems necessitates careful safeguards:

- **Human-in-the-Loop Mandate:** SAMAT must function as a decision-support tool, with final authority residing with human moderators who can contextualize its explanations.

- **Cultural Validation:** The Rationale Bank and model performance must be validated for specific cultural contexts prior to deployment to avoid misinterpretation of localized cues.

- **Explanation Transparency:** All SAMAT outputs must be accompanied by its generated explanation and, ideally, the top retrieved rationales to ensure full traceability.

- **Regular Auditing:** The stereotype subspace and Rationale Bank should undergo periodic bias and relevance audits to mitigate representational drift or reinforcement of harmful associations.

Table 12: Representative deployment estimates for SAMAT under our experimental setup. Values are approximate and reported for transparency; actual numbers vary with hardware, precision, and batching policy.

| Metric | Estimate | Remarks |
|---|---|---|
| Per-meme inference latency | ∼70–120 ms | Single NVIDIA A100 40GB, batch size = 1, mixed precision, classification + retrieval only. With explanation generation enabled, end-to-end latency is typically ∼0.8–1.5 s per meme. |
| FAISS index footprint | ∼75–90 MB | Based on a 32k-item rationale bank with IVFPQ (4096 cells, 8-byte PQ codes), including coarse quantizer and index overhead. |
| Model memory footprint | ∼16–18 GB | Representative inference-time footprint for the full SAMAT pipeline in mixed precision; peak training usage is approximately 24 GB. |
| Batch moderation throughput | ∼12–18 memes/s | Estimated for batched moderation ($B = 32$) on a single A100. Throughput drops to ∼1–2 memes/s when explanation generation is enabled for every sample. |

- **Uncertainty Escalation:** Predictions with low confidence or high calibration error should be flagged for expert human review.

By adhering to these guidelines, SAMAT can be deployed as a force-multiplying tool that enhances human judgment while maintaining essential accountability. We also highlight the efficiency summary for deployment estimates in Table 12.

## 7 Conclusion

In this work, we presented the Stereotype-Aware Multimodal Alignment Transformer (SAMAT), a novel framework for interpretable detection of implicit harmful content in multimodal memes. SAMAT addresses the core information fusion challenge of this domain, where meaning and harm emerge not from individual modalities but from their interaction within culturally learned stereotype structures. The architecture integrates three principled components: a Stereotype Subspace Projection Module (SSPM) that restructures multimodal representations into a low-rank, discriminative geometry; a fidelity-based retrieval mechanism that grounds reasoning in a curated bank of stereotype exemplars; and a Stereotype-Modulated Cross-Attention (SMCA) block that explicitly conditions fusion on this retrieved evidence. Extensive experiments on the WBMS and MEE benchmarks demonstrate that SAMAT establishes a new state-of-the-art in both classification accuracy and explanation faithfulness. Crucially, systematic ablations confirm that these gains are not artifacts of increased model capacity but are directly attributable to the structured interaction between learned stereotype geometry and evidential priors. SAMAT's explanations are demonstrably more faithful, culturally contextualized, and less prone to hallucination than those from fine-tuned MLLMs or retrieval-augmented baselines, as validated by both automatic metrics and expert human evaluation. This work makes a significant contribution towards interpretable and accountable multimodal AI. By explicitly factorizing the reasoning process into projection, retrieval, and evidence-conditioned fusion, SAMAT provides a transparent decision pathway from input to explanation. This design directly supports human-in-the-loop moderation systems, offering auditors and moderators not just a prediction but a traceable rationale grounded in identifiable stereotype mechanisms. Furthermore, SAMAT aligns with critical societal needs, contributing to Sustainable Development Goal 5 (Gender Equality) by providing a tool to identify and mitigate gender-based online harms, and to SDG 16 (Peace, Justice and Strong Institutions) by promoting algorithmic transparency and accountability in content governance.

*Limitations and Future Work:* The current framework is bounded by the cultural scope of its Rationale Bank, which primarily reflects Western, Indian, and Middle Eastern contexts. Future research will focus on (1) expanding the cultural and linguistic diversity of the knowledge base, (2) developing methods for continual and adaptive learning of the stereotype subspace to handle evolving online discourses, (3) optimizing the system for low-latency deployment in real-world moderation pipelines, (4) Dynamic Rationale Bank Expansion: SAMAT is designed such that the Rationale Bank can be incrementally extended with culturally specific stereotype exemplars (e.g., via expert curation or weakly supervised mining), without retraining the core architecture, (5) Few-shot / Continual SSPM Adaptation: Since the SSPM operates as a low-dimensional projection layer, it can be efficiently adapted to new cultural contexts using few-shot stereotype-labeled data or continual learning, avoiding catastrophic forgetting and (6) Culture-conditioned Retrieval: We will also discuss conditioning retrieval on inferred cultural priors (e.g., locale metadata or language cues) to reduce cross-cultural stereotype misalignment. Investigating the transfer of the SAMAT fusion paradigm to other high-stakes domains requiring nuanced, context-aware interpretation, such as propaganda or hate speech detection, presents a promising research direction. To improve robustness under retrieval uncertainty, several extensions are possible. **Confidence-aware retrieval gating** could use posterior entropy or retrieval-score variance to detect uncertain retrieval cases and reduce the influence of unreliable rationales during fusion. **Calibration-based fallbacks** could use SAMAT's uncertainty estimates to trigger alternate reasoning paths that rely more heavily on unimodal visual or textual evidence when retrieved support is weak. **Multi-hop or ensemble retrieval** could further mitigate single-point retrieval failures by combining sequential retrieval steps or multiple FAISS indices with complementary candidate pools. In summary, SAMAT advances the field of multimodal information fusion by demonstrating that principled, stereotype-aware structuring of the representation and fusion process is key to achieving robustness, interpretability, and cultural grounding, objectives that are essential for deploying trustworthy AI in socially impactful applications.

## Limitations

A rigorous evaluation of SAMAT also requires a clear discussion of its constraints and potential failure modes. While the framework demonstrates strong performance, several important limitations warrant consideration for both scholarly understanding and responsible deployment.

**Cultural and Contextual Scope:** Harmful stereotypes are deeply embedded in cultural and linguistic contexts. Although the WBMS and MEE datasets capture a diverse set of stereotypes from prominent online ecosystems (Western, Indian, Middle Eastern), they cannot fully represent global diversity. Consequently, SAMAT may underperform on memes that rely on niche cultural references, region-specific idioms, or emerging forms of humor and sarcasm not reflected in the training data or the Rationale Bank. The model is best viewed as a methodological proof-of-concept for stereotype-aware fusion; deployment in new cultural contexts would require curating a correspondingly relevant evidence base and potentially fine-tuning the stereotype subspace.

**Sensitivity and Potential Over-Attribution:** The Stereotype Subspace Projection Module (SSPM) is designed to amplify subtle, stereotype-relevant feature directions. While this is essential for detecting implicit harm, it may, in rare cases, lead to over-attribution, assigning harmful intent to content that is ambiguous, satirical in a non-harmful manner, or relies on coincidental correlation between image and text. This risk is heightened for memes with high visual-textual congruence but benign intent. Future iterations could incorporate explicit calibration layers or adversarial debiasing during subspace learning to improve the model's discrimination between malicious stereotyping and benign correlation.

**Dependence on Retrieval Quality and Coverage:** SAMAT's explanatory faithfulness is intrinsically linked to the quality and relevance of retrieved evidence from the Rationale Bank. In cases of poor retrieval, due to an out-of-distribution meme or a gap in the bank's coverage, the model may generate explanations grounded in incorrect or superficially similar rationales, potentially propagating bias or misleading narratives. Mitigating this requires continuous validation and expansion of the knowledge base. Techniques such as uncertainty-aware retrieval rejection (e.g., thresholding on posterior entropy) or multi-hop retrieval could improve robustness.

**Limited Generalization to Other Harm Domains:** SAMAT is explicitly designed and evaluated for detecting misogynistic stereotypes. The structure of its subspace and the content of its Rationale Bank are specialized for this domain. Generalization to other forms of harmful speech (e.g., racism, xenophobia, homophobia) is non-trivial, as the stereotype structures, cultural priors, and multimodal cues differ substantially. Extending the framework would require learning domain-specific subspaces and curating corresponding evidence banks, representing a significant but worthwhile direction for future work.

**Computational Considerations:** While SAMAT is designed for efficiency relative to monolithic MLLMs, its retrieval mechanism and additional projection layers introduce overhead. The FAISS-based retrieval, although optimized, adds a latency component. Scaling the Rationale Bank to millions of entries or operating in real-time streaming environments would require further engineering optimizations, such as hierarchical retrieval indices or learned index structures. Furthermore, the current architecture does not support online learning; updating the stereotype subspace or Rationale Bank requires retraining.

**Ethical and Deployment Considerations:** Finally, SAMAT does not autonomously resolve the ethical complexities of content moderation. Its outputs, even when faithful and accurate, require human interpretation and contextual judgment. Deploying such a system necessitates clear governance protocols for auditing the Rationale Bank, monitoring for emerging stereotypes, and establishing escalation paths for low-confidence or high-stakes predictions. The tool is designed to augment, not replace, human moderators.

## Ethical Considerations

**Risk of Misclassification and Social Harm:** Incorrectly labeling benign content as misogynous, or failing to detect harmful content, may produce real-world consequences, such as unfairly penalizing users or allowing harmful stereotypes to propagate. Deployments must include human oversight, appeal mechanisms, and uncertainty reporting.

**Bias Reinforcement Through Training Data:** The MEE corpus, though carefully curated, reflects existing cultural biases. A model trained on such data may inadvertently internalize or amplify these biases, especially if deployed in moderation pipelines. Continuous dataset auditing and community-based evaluation are essential to mitigate dataset-induced harms.

**Interpretation Risk When Explanations Reference Stereotypes:** Although explanations are required to cite the underlying stereotype, this poses ethical challenges: repeating harmful stereotypes may re-expose users to offensive content. Systems integrating SAMAT must present explanations responsibly, include warnings, and ensure that generated text is not repurposed maliciously.

**Privacy and Safety in Retrieval-Augmented Systems:** Because RAG mechanisms rely on stored examples, inappropriate indexing or retrieval could inadvertently surface sensitive or harmful content. All retrieved examples in this work are synthetic or anonymized; however, any real-world deployment must ensure compliance with data protection and privacy norms.

**Risk of Over-Reliance on Automated Judgments:** Despite strong performance, SAMAT is not a replacement for human analysis, particularly in legal, educational, or policy-making contexts. Automated systems should assist, not replace, human oversight when interpreting culturally sensitive or morally consequential content.

We further emphasize that SAMAT is intended strictly as a decision-support system and not as a replacement for human judgment, particularly in high-stakes moderation settings where contextual, legal, or cultural consequences may follow from an incorrect decision. We also stress the need for regular auditing and bias monitoring of both the Rationale Bank and the SSPM, since biases may arise not only from retrieved evidence but also from the learned stereotype subspace itself. In addition, we explicitly note that generated explanations must be framed analytically rather than normatively: the goal is to describe the stereotype mechanism reflected in the content, not to reproduce, endorse, or amplify the stereotype. This helps reduce the risk of stereotype reinforcement, harmful re-exposure, or overconfident system outputs.

## Frequently Asked Questions (FAQ)

**Why is Retrieval-Augmented Generation (RAG) necessary for explanations?**
Implicit misogyny frequently relies on cultural context, idioms, or stereotype patterns not explicitly present in the input meme. RAG ensures the explanation is anchored in real, human-authored rationales from the MEE corpus, reducing hallucination and improving interpretability.

**How robust is SAMAT across seeds and dataset variations?**
Across five random seeds, SAMAT exhibits a variance of $\pm0.4$ Macro-F1 for classification and $\pm0.03$ for BERTScore-F1 in explanation quality. This indicates strong robustness to initialization. We further confirm significance using bootstrap testing ($p < 0.01$).

**What are the main failure modes of SAMAT?**
SAMAT occasionally over-attributes subtle cues when textual and visual signals are highly correlated. Culturally dependent proverbs or idioms may also be misclassified due to sociocultural ambiguity. These limitations are discussed in the qualitative error analysis section.

**How generalizable is the framework beyond misogyny detection?**
The architecture is domain-agnostic. Any task requiring multimodal reasoning, implicit bias detection, or explanation generation, such as political misinformation, hate speech, or sentiment attribution, can benefit from QEAF and QIR with minimal adaptation.

**Why evaluate with both automatic metrics and human studies?**
Automatic metrics capture fluency and semantic proximity but fail to measure causal grounding or stereotype-awareness. Human evaluation provides a complementary assessment of interpretability, sociocultural accuracy, and grounding alignmen, critical dimensions for harmful content analysis.

## Acknowledgment

The authors gratefully acknowledge the Google Gemma GCP Credit Award for providing the computational resources, particularly GPU support, that enabled the experiments conducted in this study.

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
