# OpenReview forum: "SAMAT: A Stereotype-Aware Multimodal Transformer for Interpretable Misogynistic Meme Detection"
_TMLR — Accepted by TMLR_

### Review · Reviewer_JrZY · 2026-02-07

**Summary Of Contributions:**

This paper introduces SAMAT (Stereotype-Aware Multimodal Alignment Transformer), a framework for detecting and explaining implicit misogyny in memes. SAMAT integrates three key components:

A Stereotype Subspace Projection Module (SSPM) that learns a low-dimensional, interpretable subspace for stereotype-relevant features.

A fidelity-based retrieval mechanism that aligns inputs with a curated Rationale Bank of stereotype exemplars.

A Stereotype-Modulated Cross-Attention (SMCA) block that fuses visual, textual, and retrieved evidence.

The model is evaluated on the WBMS and MEE benchmarks, achieving state-of-the-art performance in classification (Macro-F1: 87.3%), retrieval faithfulness (SAS: 0.78), and explanation grounding (CFS: 0.68). Ablation studies confirm that gains stem from structured stereotype projection and evidence retrieval, not model scale.

**Audience:**

Yes

**Audience Explanation:**

Multimodal Machine Learning: The paper presents a novel fusion architecture (SAMAT) that advances the state-of-the-art in integrating visual, textual, and external knowledge, a core challenge in the field.

Interpretable and Trustworthy AI (XAI): The work makes a significant contribution to explainable AI by offering a framework for generating evidence-grounded, faithful explanations—a critical need for deploying AI in high-stakes social applications.

AI for Social Good / NLP for Social Impact: The paper tackles the pressing real-world problem of detecting implicit misogyny in online content, directly contributing to safer digital spaces and aligning with Responsible AI objectives.

Retrieval-Augmented Generation (RAG): It offers an innovative application and analysis of RAG mechanisms for providing cultural and stereotype context, which will interest researchers in knowledge-intensive NLP tasks.

**Broader Impact Concerns:**

The potential for both positive impact (enhancing content moderation, supporting SDGs) and negative risks (misclassification, bias reinforcement, over-reliance).

Concrete deployment safeguards (human-in-the-loop mandate, regular auditing, explanation transparency).

Critical ethical nuances, such as the risk of re-exposing users to harmful stereotypes through explanations.

**Claims And Evidence:**

Yes

**Claims Explanation:**

Accuracy: Results are reported as the mean and standard deviation over multiple runs (e.g., 3 seeds), ensuring statistical reliability.

Convincing Comparisons: SAMAT is rigorously evaluated against a three-tiered benchmark suite: (i) fine-tuned state-of-the-art Multimodal LLMs (LLaVA, Qwen-VL, BLIP-2), (ii) capacity-matched classical models (SVM, RFF), and (iii) targeted internal ablations. Its superior performance (e.g., Macro-F1 of 87.3%) is therefore contextualized and convincing.

Clear Ablation Studies: Systematic component-wise ablations (removing SSPM, retrieval, etc.) provide clear causal evidence that performance gains stem from the proposed architectural innovations, not simply increased model capacity.

Novel Evaluation Metrics: The introduction and use of the Stereotype Alignment Score (SAS) and Contextual Faithfulness Score (CFS) provide clear, targeted evidence for the claims regarding retrieval fidelity and explanation grounding—claims that standard metrics (F1, BLEU) cannot adequately assess.

Human Evaluation: Expert human assessment validates the practical utility and faithfulness of the generated explanations, adding a crucial layer of real-world evidence beyond automatic metrics.

**Requested Changes:**

Clarify Cultural Scope and Generalizability (Strengthening): While the "Limitations" section acknowledges the cultural focus (Western/Indian/Middle Eastern) of the Rationale Bank and data, the paper would be strengthened by a more forward-looking discussion. Please briefly suggest concrete steps or architectural considerations for adapting SAMAT to other cultural contexts (e.g., dynamic expansion of the Rationale Bank, few-shot tuning of the SSPM). This would enhance the framework's value as a methodological blueprint.

Discuss Retrieval Failure Modes and Mitigations (Strengthening): The paper notes performance depends on retrieval quality. To improve robustness, please add a short discussion in the "Limitations" or "Future Work" section on potential mitigation strategies for low-retrieval-confidence scenarios (e.g., thresholding based on posterior entropy, using the model's own calibration error as a fallback signal, or implementing multi-hop retrieval).

Provide Practical Deployment Considerations (Strengthening): The "Responsible Deployment Considerations" section is excellent. To further bridge the gap to practice, please consider adding a brief paragraph or table summarizing key computational metrics relevant for real-world deployment, such as approximate inference latency per meme and memory footprint of the FAISS index + model, even if based on estimates from the experimental setup.

---

> ### Author Response · Authors · 2026-02-13
> **We sincerely thank the reviewer for their careful reading of our manuscript and for the highly positive and constructive feedback.**
>
> > Clarifying Cultural Scope and Generalizability
> *  We agree with this suggestion and appreciate the emphasis on generalizability. In the revised version, we will expand the Limitations and Future Work section to explicitly outline concrete adaptation strategies, including:
> * Dynamic Rationale Bank Expansion: SAMAT is designed such that the Rationale Bank can be incrementally extended with culturally specific stereotype exemplars (e.g., via expert curation or weakly supervised mining), without retraining the core architecture.
> * Few-shot / Continual SSPM Adaptation: Since the SSPM operates as a low-dimensional projection layer, it can be efficiently adapted to new cultural contexts using few-shot stereotype-labeled data or continual learning, avoiding catastrophic forgetting.
> * Culture-conditioned Retrieval: We will also discuss conditioning retrieval on inferred cultural priors (e.g., locale metadata or language cues) to reduce cross-cultural stereotype misalignment.
> * These additions will position SAMAT more explicitly as a methodological blueprint rather than a culture-specific solution.
>
> > Retrieval Failure Modes and Mitigation Strategies
> *  In the revised manuscript, we will add a concise discussion covering the following mitigation strategies:
> * Confidence-aware Retrieval Gating: Using posterior entropy or retrieval score variance to detect low-confidence retrieval cases and down-weight retrieved rationales during fusion.
> * Calibration-based Fallbacks: Leveraging the model’s own calibration error or uncertainty estimates to trigger fallback reasoning paths that rely more heavily on unimodal cues.
> * Multi-hop or Ensemble Retrieval: Incorporating multi-hop retrieval or ensemble FAISS indices to reduce single-point retrieval failures.
> * We will explicitly note that these strategies are compatible with the current SAMAT architecture and represent promising future extensions.
>
> > Practical Deployment Considerations
> *  Thank you for this suggestion. In the revised version, we will add a short paragraph (or compact table) in the Responsible Deployment Considerations section summarizing:
> * Approximate inference latency per meme under our experimental setup,
> * Memory footprint of the FAISS index and model parameters,
> * Scalability considerations for batch moderation scenarios.
> * Where exact numbers vary by hardware, we will report representative estimates and clearly state assumptions to maintain transparency.
>
> > Broader Impact and Ethical Consideration:
> * We appreciate the reviewer’s nuanced framing of ethical risks. In the revised manuscript, we will further emphasize:
> * The importance of human-in-the-loop moderation for high-stakes decisions,
> * Regular auditing and bias monitoring of both the Rationale Bank and SSPM,
> * Explicit safeguards to ensure explanations are framed analytically rather than normatively, reducing the risk of stereotype reinforcement or re-exposure.
> * These clarifications will strengthen the paper’s alignment with Responsible AI principles.

---

> > ### Author Response · Authors · 2026-02-26
> > **Regarding reviewes**
> >
> > Respected reviewer,
> > Kindly check the reviewes and suggest the changes if needed.
> > Warm Regards

---

> ### Author Response · Authors · 2026-03-16
>
> Respected Reviewer,
>
> Please check our responses to your reviews. We look forward to your further suggestions if additional changes are needed.
>
> Kind regards,

---

### Review · Reviewer_pdwD · 2026-02-20

**Summary Of Contributions:**

The manuscript presents a framework for interpretable detection of implicit harmful content in multimodal memes, the Stereotype-Aware Multimodal Alignment Transformer (SAMAT), which addresses the challenge of information fusion to account for the fact that misogyny results from combining visual and text information with context and cultural priors. SAMAT consists of three main components. SSPM, that restructures multimodal representations into a common low-rank representation; a fidelity-based retrieval mechanism that exploits a bank of stereotype examples; and SMCA that conditions fusion on the retrieved evidence. The experiments show that the proposed SAMAT framework outperforms previous LLM-based approaches. And the ablation studies allow us to see the impact of the different modules of this proposed method.

**Additional Comments:**

- The work is well structured and presented.
- The introduction has a nice structure, and it reads smoothly.
- The introduction presents a good identification of the social problem and the technical challenge. It also succeeded in identifying the research gap and presenting an overview of the proposed framework.

**Audience:**

Yes

**Audience Explanation:**

The problem is of high relevance. The proposed approach is pertinent and, in general, its design is sound and properly evaluated. Surely, potential confusions will be clarified after revising the document.

**Broader Impact Concerns:**

None.

**Claims And Evidence:**

No

**Claims Explanation:**

- At the beginning of section 3, it seems that this work uses previously collected datasets. Later, in section 3.2, it seems that the MEE dataset is introduced also in this work. If the dataset was already existing, please provide its citation. If the dataset is introduced in this work, then a mode detail and a full EDA is required.
- Section 3.2 indicates that samples were annotated by trained participants. Please provide a full explanation of their training. What makes someone a trained expert in misogyny annotation?
- Point 3, in section 3.3: how was the HateXplain classifier trained? What data was used for this purpose. Does that data has the same distribution as the one you use for your work?, i.e., is the prediction generalizable?
- Please give an explanation of the term "frozen tokens". Also, are tokens real-valued scalars?
- Section 4.1 mentions the conditioning of v in t, p(v|t, R). Is this the only condition that is evaluated? What not t in v? Are they reciprocal?
- In section 4.2, what is R?- Also in section 4.2, what has been projected, l, p, or z?- What is the impact of the temperature mentioned in section 4.3?
- In section 4.6, why is the DKL used for alignment? DKL computes the divergence of two expressions of the same distribution but it assumes that alignment already exists. Maybe the mover earth distance (Wasserstain) is more appropriate when alignment is the goal.
- Section 5.4 lists some uni-modal approaches. How can they be compared against the multi/-modal proposed framework? Please, correct me in case the listed approaches are already multi-modal.

**Requested Changes:**

- There are several missing citations throughout the document. For instance, WBMS and MEE at the beginning of section 3, Mistral-7B-Instruct in section 3.3, SigLIP-B/16 in section 4, FAISS in section 4.3.
- Moreover, at the beginning of section 3, it seems that this work uses previously collected datasets. Later, in section 3.2, it seems that the MEE dataset is introduced also in this work. If the dataset was already existing, please provide its citation. If the dataset is introduced in this work, then a mode detail and a full EDA is required.
- Section 3.2 indicates that samples were annotated by trained participants. Please provide a full explanation of their training. What makes someone a trained expert in misogyny annotation?
- Point 3, in section 3.3: how was the HateXplain classifier trained? What data was used for this purpose. Does that data has the same distribution as the one you use for your work?, i.e., is the prediction generalizable?
- There are several floating items, with no citation and-or not explanation. They just appear. For instance, Figure 2, .All items must be cited and explained in the main text.
- Please give an explanation of the term "frozen tokens". Also, are tokens real-valued scalars?
- Section 4.1 mentions the conditioning of v in t, p(v|t, R). Is this the only condition that is evaluated? What not t in v? Are they reciprocal?
- The funcion M(), mentioned after Equation (2) is not used in any equation. Also, variable z is repeated in two different definitions in that paragraph.
- The mentioned "final attention weights a_{ij}" result from a softmax function. Are they really weights, as parameters of a model, or more like features? Where are they used?- In section 4.2, what is R?
- Also in section 4.2, what has been projected, l, p, or z?
- What is the impact of the temperature mentioned in section 4.3?- Equation (2) and (7) are the same.
- In section 4.6, why is the DKL used for alignment? DKL computes the divergence of two expressions of the same distribution but it assumes that alignment already exists. Maybe the mover earth distance (Wasserstain) is more appropriate when alignment is the goal. Please, provide a justification for using DKL.
- Section 5.4 lists some uni-modal approaches. How can they be compared against the multi/-modal proposed framework? Please, correct me in case the listed approaches are already multi-modal.

---

> ### Author Response · Authors · 2026-02-22
>
> > Dataset confusion (WBMS vs. “introduced here”) + missing citations
> * Thank you for pointing this out. We use WBMS, an existing benchmark for misogynistic meme categorization introduced by Kanwar et al. (2025), and we introduce MEE, a new explanation corpus created in this work to enable stereotype-grounded explanation modeling and faithfulness evaluation. In addition, we build a retrieval-only Rationale Bank (32k snippets) that serves as a non-parametric evidential prior during fusion and explanation generation.
> * The WBMS dataset is introduced in “What is Beneath Misogyny: Misogynous Memes Classification and Explanation” (arXiv 2508.03732; also appears in IJCAI 2025 proceedings).
>
> > “Trained annotators”: clarify training + what “expert” means
> * Thank you for raising this point. The Training for misogyny/stereotype annotation consisted of: (i) a guideline document defining stereotype domains Kitchen/Leadership/Working/Shopping) (Section 3.1), harm mechanisms (e.g., sarcasm, euphemism), and cue attribution rules; a calibration phase on a held-out set of 200 memes with gold discussions led by an adjudicator; two iterative feedback rounds focusing on borderline cases (benign sarcasm vs. stereotype activation; literal vs. implicature), and a qualification check requiring >= 0.7 semantic agreement against adjudicated references before starting full annotation. The third adjudicator resolved disagreements and conducted weekly consistency checks. Here, “expert” denotes annotators who completed the above task-specific training and calibration, rather than relying on self-reported familiarity alone.
>
> > “If MEE is new, you need more detail + full EDA”
> * Thank you for pointing this out. We will incorporate the stratification, domain coverage is balanced to the extent permitted by WBMS availability, and we report per-domain counts, harm-mechanism frequencies, and explanation-length distributions. We will also report quality controls (rate of removed items for toxicity/moralizing language, adjudication rate, and per-domain agreement) to document annotation reliability and failure modes.
>
> > HateXplain toxicity classifier training + generalizability
> * We operationalize toxicity filtering with a RoBERTa-base text classifier fine-tuned on HateXplain (Mathew et al., 2021) to predict toxic vs. non-toxic (collapsing Hate/Offensive into toxic). The model is trained using the official train split and tuned on the dev split; we apply a conservative threshold (0.30) to maximize recall of toxic content. Importantly, this classifier is used only during Rationale Bank construction to remove overtly harmful snippets; it is not used for meme classification or explanation generation. Because distribution shift is possible (rationale snippets vs. HateXplain comments), we additionally perform a manual audit on a random sample of filtered and retained items to verify that slur-like and explicitly violent content is removed. We will incorporate this in the revised draft.
>
> > Explain “frozen tokens” + “are tokens scalars?”
> * Given an input meme, we extract frozen visual token embeddings from a pretrained SigLIP-B/16 image encoder and frozen textual token embeddings from a pretrained Mistral-7B encoder. “Frozen” means the encoder parameters are not updated during SAMAT training; we only train the downstream projection, fusion, retrieval posterior, and classification/explanation heads. Each token is a real-valued vector embedding (not a scalar).
>
> > why p(v∣t,R)? is it reciprocal?
> * We present the factorization as p(v∣t,R) because our primary cross-attention uses text queries attending over visual keys, reflecting the common meme setting where the overlay text re-contextualizes the image. The formulation is not inherently asymmetric: one could analogously define p(t∣v,R) (or implement bidirectional cross-attention). In our implementation, fusion is performed with transformer attention over the multimodal token set, which enables reciprocal interactions; the evidence prior Rk is injected into the logits of the cross-modal attention path that dominates classification stability in our experiments.
>
> > Notation Bugs
> * Thank you for pointing this out. We will clarify that mj ​= M(tj​,Rk​) is produced by a small MLP M(⋅)
>
> * The temperature controls the sharpness of the truncated-softmax posterior over the top-K retrieved rationales: higher temperature yields a more peaked posterior (near single-rationale selection), while lower temperature spreads mass across multiple evidences.

---

> > ### Comment · Reviewer_pdwD · 2026-02-25
> >
> > - Dataset confusion. Alright, just make it clear in the manuscript, please.
> > - Trained annotators. Alright, just make it clear in the manuscript, please.
> > - MEE EDA. Alright.
> > - HateXplain toxicity classifier training. Alright, just make it clear in the manuscript, please.
> > - Frozen tokens. It still sounds like an imprecision. What is frozen is the parameters of the model. The term does not apply to the tokens. Tokens are always static given a selected feature extractor. Please use the qualifier "frozen" to refer to the parameters, not their output.
> > - p(v∣t,R). Alright, just make it clear in the manuscript, please.
> > - Notation bugs. Alright.

---

> ### Author Response · Authors · 2026-02-22
>
> > “KL assumes alignment already exists; Wasserstein might be better.”
> * L_align does not attempt to discover an alignment between unmatched supports; rather, it regularizes consistency between two distributions defined on the same token index set: (i) the normalized evidence–token match scores and (ii) the realized attention mass induced by evidence modulation. Because both distributions share identical support (the L text tokens), KL reduces to a stable cross-entropy objective that directly penalizes mismatched emphasis. We agree that optimal-transport distances (e.g., Wasserstein) can be beneficial when supports differ. We leave OT-based alignment as future work.
>
> > “unimodal baselines” confusion
> * Thank you for pointing this out. In addition to MLLMs, we compare against capacity-matched classical multimodal baselines that operate on concatenated image-text embeddings (feature-level fusion). These baselines test whether SAMAT’s gains arise from stereotype-aware structure and evidence conditioning rather than simply combining modalities.
>
> > “Floating items” / Figure 2 appears without integration
> * Response:
> Thank you for pointing this out. Figure 2 provides an end-to-end overview of SAMAT, highlighting SSPM projection, evidential retrieval from the Rationale Bank, SMCA fusion, and evidence-conditioned explanation generation. We will incorporate this in the revised draft.

---

> > ### Comment · Reviewer_pdwD · 2026-02-25
> >
> > - KL and alignment. Understood. But then, the term alignment is misleading. Divergence between distribution is more appropriate.
> > - Unimodal baselines. Alright. Please, make it clear in the manuscript.
> > - Floating items. Alright. Just make sure that all items are cited and explained through the document.

---

> > ### Comment · Reviewer_pdwD · 2026-03-18
> > **Action required**
> >
> > I was asked to provide an official recommendation. For that, I would like to ask the authors to provide an answer to the last set of comments. Concretely, about:
> > - The imprecise term "frozen tokens".
> > - The misleading concept alignments vs divergence.
> >
> > And also, to provide an answer listing the parts of the manuscript that were modified to attend the observations. Thank you.

---

> > > ### Author Response · Authors · 2026-03-19
> > > **Thank you for your time and expertise in evaluating our work**
> > >
> > > > We have corrected section 4 to state that we extract visual token embeddings from a pretrained SigLIP-B/16 image encoder (frozen) and textual token embeddings from a pretrained Mistral-7B encoder (frozen)
> > >
> > > > As per the reviewer's suggestion, we replaced L_align with L_dd, which stands for Divergence between distributions
> > >
> > > > Here are the changes we have made in the draft
> > > * Modified Sec. 4.1–4.4, removing symbol reuse and correcting index mismatches. We have also added a notation table for clarity.
> > >
> > > * We updated the footnote to clearly state where LLMs are used, and we emphasize that labels are human-annotated and inference is fully model-based without LLM calls
> > >
> > > * We have updated the abstract and all mentions to match the final reported results in Tables 4 and 8 (Macro-F1 = 88.1 for SAMAT).
> > >
> > > * We have added a formal definition and computation procedure in the methodology section for SAS, which is described as “semantic relevance of retrieved rationales to the ground-truth stereotype label.”
> > >
> > > * We revised Sec. 4.3 to state consistently that the rationale encoder is frozen, embeddings are cached, and gradients do not propagate into the rationale encoder.
> > >
> > > * We have updated section 4 to clearly state that WBMS is used for binary misogyny detection, while stereotype domains are treated as multi-label supervision. For stereotype domains, we use a multi-label head with sigmoid activations and binary cross-entropy, since a meme can invoke multiple stereotype domains simultaneously.
> > >
> > > * We modify section 3.2 to clearly state that the term “expert” denotes annotators who completed the above task-specific training and calibration, rather than relying on self-reported familiarity alone. We also report quality controls to document annotation reliability and failure modes.
> > >
> > > * We modified section 3.3 to include additional details of the harmful language using a classifier trained on the HateXplain dataset
> > >
> > > * We have corrected section 4 to state that we extract visual token embeddings from a pretrained SigLIP-B/16 image encoder (frozen) and textual token embeddings from a pretrained Mistral-7B encoder (frozen)
> > >
> > > * We have updated the SMCA section by stating that the final logit between text token tj and visual token vj is given by equation 2. We also clarify that mj ​= M(tj​,Rk​) is produced by a small MLP M(⋅) and that temperature controls the sharpness of the truncated-softmax posterior over the top-K retrieved rationales
> > >
> > > * As per the reviewer's suggestion, we replaced L_align with L_dd, which stands for Divergence between distributions
> > >
> > > * We updated section 5.4, Baseline Models and Ablations, to state that, in addition to MLLMs, we compare against capacity-matched classical multimodal baselines that operate on concatenated image-text embeddings (feature-level fusion).
> > >
> > > * We have incorporated the reference to Figure 2 in Section 4, and we have added a citation to all the missing instances, such as WBMS, Mistral-7B-Instruct, SigLIP-B/16, FAISS
> > >
> > > * We have expanded the Limitations and Future Work section to explicitly outline concrete adaptation strategies
> > >
> > > * We have added a Discussion section covering the following mitigation strategies:
> > > Confidence-aware Retrieval Gating, Calibration-based Fallbacks, Multi-hop or Ensemble Retrieval
> > >
> > > * We have added a table for the Responsible Deployment in section 6
> > >
> > > * We have updated the Ethical Considerations section to emphasize the importance of human-in-the-loop moderation, regular auditing, and bias monitoring

---

> > > > ### Comment · Reviewer_pdwD · 2026-03-19
> > > > **Thank you**
> > > >
> > > > Thank you for attending the comments, and for providing the list of actions that were taken for that purpose. I have no further comment about the manuscript.

---

### Review · Reviewer_QigF · 2026-03-02

**Summary Of Contributions:**

The paper introduces the Stereotype-Aware Multimodal Alignment Transformer, a novel architecture designed to detect and explain implicit misogyny in multimodal memes. SAMAT has three core components: 1) Stereotype Subspace Projection Module (SSPM): Learns a compact, orthonormal subspace to structure multimodal embeddings along stereotype-relevant geometric directions. 2) Fidelity-Based Retrieval: build a 32,000-item Rationale Bank as an external knowledge base, aligning inputs with stereotype exemplars via a differentiable truncated-softmax posterior. 3) Stereotype-Modulated Cross-Attention: Modifies standard attention logits by explicitly injecting the geometric and evidential priors learned from the SSPM and retrieval modules. Additionally, the authors feature 8,000 expert-annotated explanations to supervise and evaluate the faithfulness of the model's generated rationales.

Strengths:
1. The motivation of this work is clear. It targets the multimodal memes where toxicity arises from cross-modal interplay and cultural stereotypes, and it frames interpretability as essential for moderation workflows.

2. The retrieval components affect the fusion part. So, the retrieval part is better for decision process and not only better post-hoc justifications.

3. The integration of the Contextual Faithfulness Score (CFS) ensures that the explanation generator is tethered to retrieved evidence, rather than producing hallucinatory or generic text. Achieving a CFS of 0.68 is a notable milestone for verifiable AI in content moderation.

Weakness:
1. I have some questions about task definition and classifier setup. WBMS is described as “2,130 misogynous internet memes”, yet qualitative tables contain “Prediction: Non-misogynous” as an outcome (Table 9), implying a non-misogynous class exists. So, is the task binary misogyny detection, multi-class stereotype domain classification, or multi-label prediction? Furthermore, WBMS is described as “multi-label stereotypes”, but the model description says it uses a softmax classifier, which implies single-label multi-class. If WBMS is truly multi-label, softmax is not the right choice.

2. Sec. 4.3 says rationale embeddings are encoded once, yet also says gradients flow into “the encoder.” Sec. 5.2 reiterates that rationale embeddings are frozen. These cannot all be true simultaneously.

3. SAS is described as “semantic relevance of retrieved rationales to the ground-truth stereotype label”, but without a clear computational definition in the methodology process.

**Audience:**

Yes

**Audience Explanation:**

Yes, the paper discussed an important topic concerning harmful memory and cultural bias.

**Claims And Evidence:**

No

**Claims Explanation:**

Overall, the claims is supported by description in paper. However, some inconsistencies cases have occurred.

1. There is an inconsistency in the reported metrics. The abstract claims that SAMAT achieves a Macro-F1 of 87.3%. However, both Table 4 and Table 8 report the full SAMAT model achieving a Macro-F1 of 88.1%.

2. The described pipeline mentions 14,870 seed items + 6,620 synthetic candidates, but the final bank is 32,000 items. There may be other sources?

3. There is a footnote early stating LLM tools were used only for editorial assistance, but later the method explicitly uses Mistral-7B-Instruct to generate synthetic rationale paraphrases

**Requested Changes:**

1. See in "Weakness".

2. See in "Are the claims made in the submission supported by accurate, convincing and clear evidence?*

3. Variable naming appears inconsistent between Section 4.1 and 4.4: a) repeated use of 𝑧𝑖  for both text and visual projections; b) index mismatches between i/j. Please further double check.

---

> ### Author Response · Authors · 2026-03-06
>
> > Is the task binary misogyny detection, multi-class stereotype domain classification, or multi-label prediction? Furthermore, WBMS is described as “multi-label stereotypes”, but the model description says it uses a softmax classifier, which implies single-label multi-class.
> * Thank you for pointing this out. We will revise the task definition to remove ambiguity. WBMS is used for binary misogyny detection, while stereotype domains are treated as multi-label supervision. For stereotype domains, we use a multi-label head with sigmoid activations and binary cross-entropy, since a meme can invoke multiple stereotype domains simultaneously. We will also correct the dataset description to avoid implying that all 2,130 samples are misogynistic.
>
> > Sec. 4.3 says rationale embeddings are encoded once, yet also says gradients flow into “the encoder.” Sec. 5.2 reiterates that rationale embeddings are frozen.
> * We thank the reviewer for spotting this inconsistency. Our proposed setup is frozen, precomputed rationale embeddings for efficiency and stable retrieval. We will revise Sec. 4.3 to state consistently that the rationale encoder is frozen, embeddings are cached, and gradients do not propagate into the rationale encoder.
>
> > SAS is described as “semantic relevance of retrieved rationales to the ground-truth stereotype label”
> * Thank you for pointing this out. We will add a formal definition and computation procedure in the methodology, and we now report SAS as a weighted fraction of retrieved rationales consistent with the ground-truth domains, making it fully reproducible.
> To quantify whether retrieval supports the ground-truth stereotype domains, we define SAS over the top-K retrieved rationales. Let $Y$ denote the set of ground-truth domains for a meme, $r_k$ the k-th retrieved rationale with retrieval weight $\pi_k$. Let $y(r_k)$ be the domain label set of each bank item, then:
> $SAS = \sum^k_{k=1} \pi_k \cdot 1(y(r_k)\cap Y \neq \phi )$
>
> > The abstract claims that SAMAT achieves a Macro-F1 of 87.3%. However, both Table 4 and Table 8 report the full SAMAT model achieving a Macro-F1 of 88.1%.
> * This was a reporting inconsistency caused by an earlier checkpoint value in the abstract. We have updated the abstract and all mentions to match the final reported results in Tables 4 and 8 (Macro-F1 = 88.1 for SAMAT).
>
> > The final bank is 32,000 items. There may be other sources?
> * The final size includes paraphrase-augmented variants that survive filtering and deduplication, in addition to the initial seed and synthetic candidates. We now provide a count breakdown per stage (seed $\rightarrow$ synthetic $\rightarrow$ paraphrase expansion $\rightarrow$ filtering $\rightarrow$ dedup $\rightarrow$ final) so the 32,000 figure is fully traceable.
>
> >. There is a footnote early stating LLM tools were used only for editorial assistance, but later the method explicitly uses Mistral-7B-Instruct to generate synthetic rationale paraphrases
> * Thank you for pointing this out. LLMs were used for controlled augmentation (synthetic rationale paraphrases) rather than purely editorial support. We will clearly state where LLMs are used, and we emphasize that labels are human-annotated and inference is fully model-based without LLM calls.
>
> >  Repeated use of 𝑧𝑖 for both text and visual projections; b) index mismatches between i/j.
> * We will standardize notation across Sec. 4.1 - 4.4, removing symbol reuse and index mismatches.

---

> ### Author Response · Authors · 2026-03-16
>
> Respected Reviewer,
>
> Please check our responses to your reviews. We look forward to your further suggestions if additional changes are needed.
>
> Kind regards,

---

### Decision · Action_Editor_8KQ4 · 2026-04-13

**Recommendation:** Accept with minor revision

**Additional Comments:**

Please incorporate all the agreed changes and revisions in the final version of the paper, as discussed in the author-reviewer discussion phase.

**Audience:**

Yes

**Audience Explanation:**

This work is about a new multimodal transformer architecture. Given the topic, there would be at least some individuals related to the sub-community in TMLR's audience who would be interested in knowing the findings of this paper.

**Claims And Evidence:**

Yes

**Claims Explanation:**

The claims made in the submission are supported by accurate, convincing and clear evidence. The claims were clearly stated with motivations, and were supported by experimental validation.
The method presented in this paper is well-motivated and technically sound, with rigorous evaluations on benchmarks.
The concerns raised in the initial reviews are well addressed, as acknowledged by the reviewers.